# Position: Epistemic Uncertainty Estimation Methods are Fundamentally Incomplete

**Sebastián Jiménez** [* 1]  **Mira Jürgens** [* 1]  **Willem Waegeman** [1]

## Abstract

Identifying and disentangling sources of predictive uncertainty is essential for trustworthy supervised learning. We argue that widely used second-order methods that disentangle aleatoric and epistemic uncertainty are fundamentally incomplete. First, we show that unaccounted bias contaminates uncertainty estimates by overestimating aleatoric (data-related) uncertainty and underestimating the epistemic (model-related) counterpart, leading to incorrect uncertainty quantification. Second, we demonstrate that existing methods capture only partial contributions to the variance-driven part of epistemic uncertainty; different approaches account for different variance sources, yielding estimates that are incomplete and difficult to interpret. Together, these results highlight that current epistemic uncertainty estimates can only be used in safety-critical and high-stakes decision-making when limitations are fully understood by end users and acknowledged by AI developers.

## 1. Introduction

Uncertainty estimation in machine learning is essential for assessing the reliability of model predictions. In recent supervised learning research, one often distinguishes between two types of uncertainty: aleatoric and epistemic uncertainty (Kendall & Gal, 2017; Hüllermeier & Waegeman, 2021). Assuming that the ground truth data-generating process can be described by a conditional probability distribution $p(y|\boldsymbol{x})$, aleatoric uncertainty reflects the inherent randomness of this distribution, and is hence irreducible. Epistemic uncertainty, on the other hand, appears because

the unknown $p(y|\boldsymbol{x})$ needs to be estimated using data. As a result, epistemic uncertainty can be reduced by collecting more data, improving model specification, or changing the learning algorithm. While the correct *disentanglement* (Depeweg et al., 2018; Valdenegro-Toro & Mori, 2022) of these two types of uncertainty is crucial for model interpretability, and for downstream tasks such as out-of-distribution detection (Hendrycks & Gimpel, 2017) and active learning (Gal et al., 2017), it is far from trivial.

One principled way of representing epistemic uncertainty is through a *second-order distribution*, that is, a distribution over possible predictive distributions (Bengs et al., 2022). In this work, we focus on second-order distributions because many existing uncertainty quantification approaches can be interpreted within this framework, including Bayesian methods, ensemble methods, and deterministic approaches (Gawlikowski et al., 2023; He et al., 2025). In Bayesian models, uncertainty is represented by a posterior distribution over model weights, which induces a distribution over a classical first-order distribution (e.g. a distribution over mean and variance of a Gaussian in a regression setting). In posterior predictive sampling, one takes multiple samples of the weights from the posterior distribution and computes the corresponding second-order distribution (Gal & Ghahramani, 2016; Denker & LeCun, 1990; Neal, 2012; Blundell et al., 2015). Ensemble methods similarly induce a distribution over first-order distributions by aggregating outputs from multiple independently trained models (Lakshminarayanan et al., 2017; Ovadia et al., 2019). Finally, some approaches aim to model epistemic uncertainty more directly by learning the parameters of the second-order distribution using specific loss functions (Charpentier et al., 2022; Amini et al., 2020; Sensoy et al., 2018; Malinin & Gales, 2018). Viewed through this lens, these approaches provide a common framework for disentangling aleatoric and epistemic uncertainty.

Under a second-order representation, uncertainty can be disentangled in different ways depending on the learning task. In classification, uncertainty is commonly analyzed using information-theoretic decompositions, which separate total, aleatoric, and epistemic uncertainty (Depeweg et al., 2018). For regression, analogous variance-based decompositions

---

[*]Equal contribution [1]Department of Data Analysis and Mathematical Modeling, Ghent University, Ghent, Belgium. Correspondence to: Sebastián Jiménez <sebastian.jimenez@ugent.be>, Mira Jürgens <mira.juergens@ugent.be>.

*Proceedings of the 43$^{rd}$ International Conference on Machine Learning*, Seoul, South Korea. PMLR 306, 2026. Copyright 2026 by the author(s).

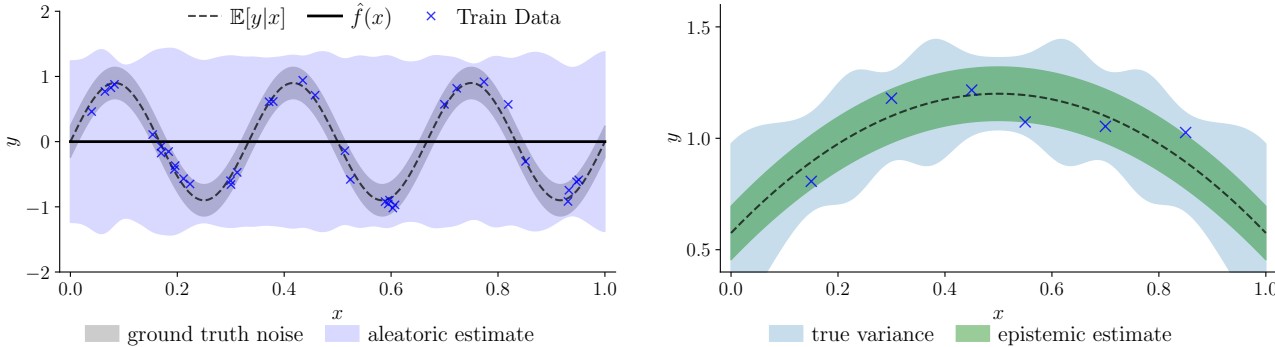

*Figure 1.* **Conceptual illustration of our two main arguments for the regression case.** Left: When high model bias is present, we show that aleatoric uncertainty *overestimates* the ground truth noise. (Sec. 3.1). Right: Many second-order methods capture only *procedural* and not data uncertainty (Sec. 3.2), thereby underestimating the variance part of the epistemic uncertainty.

are typically employed (Lakshminarayanan et al., 2017; Depeweg et al., 2018; Sale et al., 2023). However, a growing body of critical work argues that such second-order decompositions are not theoretically well-founded and can be difficult to interpret in practice. For evidential second-order risk minimization, recent analyses highlight identifiability and convergence pathologies, showing that the resulting epistemic uncertainty estimates are tightly controlled by regularization, making them only meaningful in a relative sense (Bengs et al., 2022; Meinert et al., 2023; Jürgens et al., 2024). Complementarily, in Bayesian deep learning, Fellaji & Pennerath (2024) report failure modes in which epistemic uncertainty can collapse as model size increases, contrary to what is expected. More broadly, Kirchhof et al. (2025) emphasizes that aleatoric/epistemic definitions and decomposition principles can be mutually inconsistent, and that practical estimators may become strongly correlated, challenging the premise that these components are meaningfully disentangled. In the same sense, recent works show that these decompositions are theoretically dubious (Wimmer et al., 2023; Smith et al., 2025), and may produce highly correlated aleatoric and epistemic uncertainty estimates (Mucsányi et al., 2024), questioning their ability to disentangle different sources of uncertainty meaningfully.

**Our position.** The most prominent recent papers that propose uncertainty disentanglement methods have collectively received almost 40,000 citations[1], and their usage in safety-critical applications is increasing. This relevance, together with the recent critical work motivate a more fundamental reconsideration of what epistemic uncertainty is intended to represent and quantify in predictive models. Accordingly, **we argue that recent and highly-cited epistemic uncertainty estimation methods are fundamentally incomplete**. To substantiate this claim, we proceed as follows. In Section

---

[1]Number obtained from Google Scholar. Check Appendix A for details.

2, we review and systematize the different sources of epistemic uncertainty identified in the literature, distinguishing between estimation uncertainty (variance-related), model uncertainty (bias-related), and distributional uncertainty. In Section 3.1 we adopt the bias–variance decomposition of Bregman divergences (Hastie et al., 2009; Gruber et al., 2023) and demonstrate that the bias component is as central to epistemic uncertainty as the variance component. Within this framework, we show that **bias induced by the regularization mechanisms commonly used in neural networks can substantially distort uncertainty estimates**, even when the true data-generating function lies within the model's hypothesis space. In Section 3.2, we turn to the variance component and demonstrate that *existing uncertainty quantification methods fail to capture it in its entirety*; instead, different approaches account for different variance contributions, resulting in estimates that are inherently incomplete and difficult to interpret. Finally, in Section 4 we discuss alternative perspectives on epistemic uncertainty proposed in the literature and relate them to our work.

## 2. Setup and Preliminaries

**Supervised learning setup.** We consider an instance space $\mathcal{X} \subset \mathbb{R}^D$ and label space $\mathcal{Y}$, where $\mathcal{Y} \subseteq \mathbb{R}$ for regression and $\mathcal{Y} = \{1, \dots, K\}$ for classification. We assume a dataset $\mathcal{D}_N = \{(\boldsymbol{x}_i, y_i)\}_{i=1}^N$ of i.i.d. samples from an unknown joint distribution $P$ on $\mathcal{X} \times \mathcal{Y}$, with ground-truth conditional distribution $p(y|\boldsymbol{x})$. As a loss function $\ell : \mathcal{P}(\mathcal{Y}) \times \mathcal{Y} \to \mathbb{R}$ we assume a *strictly proper scoring rule* (Gneiting & Raftery, 2007) which uniquely minimizes the expected loss when the prediction equals the ground truth. Every strictly proper scoring rule induces a convex entropy $\mathbb{H}_\ell$ and a Bregman divergence $D_\ell$ such that the pointwise expected score gap equals the divergence:

$$D_\ell(p(\cdot|\boldsymbol{x}), \hat{p}(\cdot|\boldsymbol{x})) := \mathbb{E}_{y|\boldsymbol{x}}[\ell(\hat{p}(\cdot|\boldsymbol{x}), y)] - \mathbb{E}_{y|\boldsymbol{x}}[\ell(p(\cdot|\boldsymbol{x}), y)],$$

with entropy $\mathbb{H}_\ell(p(\cdot|\boldsymbol{x})) := \mathbb{E}_{y|\boldsymbol{x}}[\ell(p(\cdot|\boldsymbol{x}), y)]$ (Banerjee et al., 2005; Gneiting & Raftery, 2007). For a hypothesis space $\mathcal{H} := \{p_{\boldsymbol{\theta}} : \mathcal{X} \to \mathcal{P}(\mathcal{Y})|\boldsymbol{\theta} \in \Theta\}$, of possible predictive distributions parametrized by $\boldsymbol{\theta} \in \Theta$, the *risk* of an estimator $\hat{p} \in \mathcal{H}$ integrates the pointwise expected loss over $\boldsymbol{x}$:

$$\mathcal{L}(\hat{p}) := \mathbb{E}_{\boldsymbol{x}}\left[\mathbb{E}_{y|\boldsymbol{x}}[\ell(\hat{p}(\cdot|\boldsymbol{x}), y)]\right] = \mathbb{E}_{(\boldsymbol{x},y)}[\ell(\hat{p}(\cdot|\boldsymbol{x}), y)].$$

By strict properness, the unconstrained risk minimizer is $p(y|\boldsymbol{x})$ itself. The *Bayes-optimal estimator* within $\mathcal{H}$ is

$$\hat{p}^* := \arg\min_{\hat{p} \in \mathcal{H}} \mathcal{L}(\hat{p}). \tag{1}$$

**Second-order distributions.** Following common literature (Bengs et al., 2022; Hüllermeier & Waegeman, 2021; Gawlikowski et al., 2023; Li et al., 2025), we represent uncertainty via a second-order distribution $q(\boldsymbol{\theta}|\boldsymbol{x})$ over parameters $\theta \in \Theta$ of the (first-order) distribution. The predictive distribution can then be obtained by model averaging:

$$\hat{p}(y|\boldsymbol{x}) = \int_\Theta p(y|\boldsymbol{\theta}, \boldsymbol{x})\, q(\boldsymbol{\theta}|\boldsymbol{x})\, d\boldsymbol{\theta}. \tag{2}$$

Existing measures that decompose the uncertainty of $q(\boldsymbol{\theta}|\boldsymbol{x})$ into an aleatoric and epistemic component focus on entropy (Depeweg et al., 2018), distance to reference sets (Sale et al., 2023), or proper scoring rules (Hofman et al., 2024; Fishkov et al., 2025). In regression, the *variance-based decomposition* (Depeweg et al., 2018; Sale et al., 2023) is most commonly used. In this approach, the variance of the prediction $\hat{y}$, obtained via model averaging as in Eqn. 2, is decomposed into an aleatoric and an epistemic part:

$$\text{Var}(\hat{y}|\boldsymbol{x}) = \underbrace{\mathbb{E}_{\boldsymbol{\theta}}\left[\text{Var}(\hat{y}|\boldsymbol{x}, \boldsymbol{\theta})\right]}_{\text{aleatoric estimate}} + \underbrace{\text{Var}_{\boldsymbol{\theta}}\left(\mathbb{E}[\hat{y}|\boldsymbol{x}, \boldsymbol{\theta}]\right)}_{\text{epistemic estimate}}. \tag{3}$$

**Sources of epistemic uncertainty.** As aleatoric uncertainty corresponds to the intrinsic randomness of the data-generating process, one can directly associate it with the stochasticity induced by the ground truth conditional distribution, using measures such as variance or entropy. Epistemic uncertainty, on the other hand, measures the uncertainty about the ground truth, i.e., the expected deviation from it. It is reducible by collecting more knowledge, hence decreasing the uncertainty about the parameters $\boldsymbol{\theta}$, but also the chosen hypothesis space $\mathcal{H}$. This uncertainty can stem from different sources, which several authors discuss (Hüllermeier & Waegeman, 2021; Huang et al., 2023; Malinin & Gales, 2018). Unifying those perspectives, we will distinguish in total five distinct sources of epistemic uncertainty (see Figure 2): approximation error, estimation bias, data uncertainty, procedural uncertainty and distributional uncertainty.

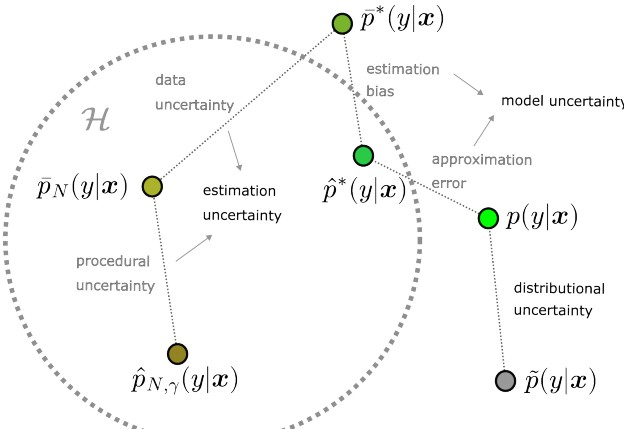

*Figure 2.* Sources of epistemic uncertainty in machine learning, where $p(y|\boldsymbol{x})$ denotes the true data-generating process. Different sources of epistemic uncertainty can lead to the estimate $\hat{p}(y|\boldsymbol{x})$ being further from the ground truth. See Sec. 2 for further details. Figure inspired from Huang et al. (2023)

Following (Huang et al., 2023), we define $\mathcal{D}_N \sim P^N$ as the random variable reflecting the *data uncertainty*, where $P^N$ is the distribution over all datasets of size $N$. Similarly, let $\gamma \sim P^\gamma$ be the random variable reflecting all *procedural uncertainty*, which arises from the training process from a single model (due to e.g. random weight initialization or data ordering) and is present even with infinite data. $P^\gamma$ denotes the distribution over which the procedural parameters are sampled. An estimator $\hat{p}_{N,\gamma}(\cdot|\boldsymbol{x})$ depends on both types of uncertainties. We make use of the Bregman centroid (Banerjee et al., 2005; Pfau, 2013)

$$\bar{p}^*(\cdot|\boldsymbol{x}) := \arg\min_{p \in \mathcal{P}(\mathcal{Y})} \mathbb{E}_{\mathcal{D}_N,\gamma}\left[D_\ell(p, \hat{p}_{N,\gamma}(\cdot|\boldsymbol{x}))\right]. \tag{4}$$

Note that $\bar{p}^*$ does not have to lie in $\mathcal{H}$, as depicted in Figure 2. For example, when $\mathcal{H}$ only consists of decision stumps, the Bregman centroid might be a more complex decision tree. One can look at two main sources of epistemic uncertainty.

**Model uncertainty.** The discrepancy between the Bregman centroid $\bar{p}^*(y|\boldsymbol{x})$ and the ground truth $p(y|\boldsymbol{x})$ can be further decomposed into *estimation bias*, the systematic deviation of $\bar{p}^*(y|\boldsymbol{x})$ from the Bayes estimator $\hat{p}^*(y|\boldsymbol{x})$ (Eq. 1) and *approximation error*, which is the difference between $\hat{p}^*(y|\boldsymbol{x})$ and $p(y|\boldsymbol{x})$ (Vapnik, 2013; Brown & Ali, 2024):

$$\begin{aligned} \mathbb{E}_{(\boldsymbol{x},y)}&[D_\ell(p(\cdot|\boldsymbol{x}), \bar{p}^*(\cdot|\boldsymbol{x}))] \\ &= \underbrace{\mathcal{L}(\hat{p}^*) - \mathcal{L}(p)}_{\text{approximation error}} + \underbrace{\mathcal{L}(\bar{p}^*) - \mathcal{L}(\hat{p}^*)}_{\text{estimation bias}}. \end{aligned} \tag{5}$$

**Estimation uncertainty.** Due to having a limited dataset and procedural uncertainty, the variation of the estimate

$\hat{p}_{N,\gamma}(y|\boldsymbol{x})$ around the Bregman centroid $\bar{p}^*(y|\boldsymbol{x})$ forms another part of epistemic uncertainty. It further decomposes into *data uncertainty* and *procedural uncertainty*. For a fixed dataset $\mathcal{D}_N = \{(\boldsymbol{x}_i, y_i)\}_{i=1}^N$, we can define the dataset-conditional centroid as

$$\bar{p}_N(\cdot|\boldsymbol{x}) = \arg\min_{p \in \mathcal{P}(\mathcal{Y})} \mathbb{E}_\gamma[\mathcal{D}_\ell(p, \hat{p}_{N,\gamma}(\cdot|\boldsymbol{x}))|\mathcal{D}_N],$$

where the expectation is taken over the procedural parameters $\gamma$. Data uncertainty can then be defined as the difference between the (global) Bregman centroid $\bar{p}^*(y|\boldsymbol{x})$ and the dataset-conditional version $\bar{p}_N(y|\boldsymbol{x})$. Similarly, procedural uncertainty can be defined as the difference between $\bar{p}_N(y|\boldsymbol{x})$ and $\hat{p}_{N,\gamma}(y|\boldsymbol{x})$. Using this notation, estimation uncertainty is the difference between $\bar{p}^*(y|\boldsymbol{x})$ and $\hat{p}_{N,\gamma}(y|\boldsymbol{x})$, as visualized in Figure 2.

**Distributional uncertainty.** Another type of uncertainty occurs if there is a distribution shift between training and test data (Malinin & Gales, 2018), respectively, and their associated conditional distributions $p(y|\boldsymbol{x})$ and $\tilde{p}(y|\boldsymbol{x})$.

# 3. Epistemic Uncertainty Estimation Methods are Fundamentally Incomplete

In this section we argue that current approaches to estimating aleatoric and epistemic uncertainty are fundamentally incomplete. To our opinion, this incompleteness has two main reasons:

- **First, estimation bias and approximation error both distort epistemic and aleatoric uncertainty estimates**. Since second-order representations do not have access to the ground truth conditional class distribution, it is not possible to incorporate model uncertainty into epistemic uncertainty estimates. Consequently, a complete representation of epistemic uncertainty is fundamentally unattainable (see Section 3.1).

- **Second, existing estimators are also incomplete because they do not represent data uncertainty and procedural uncertainty at the same time**. From a bias-variance perspective, these two types of uncertainty are closely related to the variance of predictions (see Section 3.2).

To illustrate our arguments, we will focus on the estimation of uncertainty in a regression setting. Much of the existing critical literature on uncertainty disentanglement has concentrated on the classification setting, but the regression setting allows us to present the core ideas more transparently, while remaining complementary to prior work. The bias-driven failure mode also arises in the classification setting: under the information-theoretic decomposition, systematic

model bias inflates the aleatoric component while the epistemic estimate remains low in precisely the regions where the model is least trustworthy (Appendix C.4). We consider a data-generating process such that $y_i = f(\boldsymbol{x}_i) + \epsilon_i$ and $\epsilon_i \sim \mathcal{N}(0, \sigma^2(\boldsymbol{x}_i))$. Here, $\epsilon_i$ is data-dependent (*heteroscedastic*) noise, and $f$ is a deterministic function, such that $p(\cdot|\boldsymbol{x})$ forms the density of a Gaussian distribution with mean $f(\boldsymbol{x})$ and variance $\sigma^2(\boldsymbol{x})$.

For illustrations we make use of the synthetic regression setting of Rossellini et al. (2024). A dataset $\mathcal{D}_N = \{(x_i, y_i)\}_{i=1}^N$ is generated where $x_i \sim Beta(1.2, 0.5)$ and

$$y_i = \sin\left(\frac{1}{5(x_i + 0.16)^3}\right) + \epsilon_i, \quad \epsilon_i \sim \mathcal{N}(0, \sigma^2(x_i)), \tag{6}$$

with $\sigma^2(x) = x^4$. This data-generating process is chosen because it represents a problem with two main regions. The left side of the data domain ($x < 0.4$) has low true aleatoric uncertainty and a difficult-to-learn function, which is expected to induce a high epistemic uncertainty. The right side of the data domain ($x \geq 0.4$) has high true aleatoric uncertainty and a simple function (which is expected to induce a low epistemic uncertainty). This one-dimensional setting is ideal for visually illustrating our arguments regarding uncertainty disentanglement (Figures 3 and 4). As an example, we will illustrate the behavior of Deep Ensembles, considering a simple multi-layer perceptron that outputs $\hat{f}(x)$ and $\hat{\sigma}^2(x)$ as the mean and variance of the Gaussian distribution at $x$. Predictions coming from different parametrizations of the neural network are then aggregated to obtain a second-order distribution. In Appendix C we provide similar illustrations for a real-world regression dataset and for other second-order models.

## 3.1. Bias distorts uncertainty estimations

From a statistician's viewpoint, the classical bias-variance decomposition provides further insight into the interplay between estimation uncertainty and aleatoric uncertainty (Hastie et al., 2009; Gruber et al., 2023). While it is originally defined for the mean squared error as a loss function, it can also be generalized to any proper scoring rule (Gruber & Buettner, 2023; Pfau, 2013). As a small further extension, we incorporate data and procedural components into the decomposition:

$$\mathbb{E}_{\mathcal{D}_N,\gamma}[\mathcal{L}(\hat{p})] = \underbrace{\mathbb{H}_\ell(p(y|x))}_{\text{aleatoric uncertainty}} + \underbrace{D_\ell(p(y|x), \bar{p}^*(y|x))}_{\text{generalized bias}}$$
$$+ \underbrace{\mathbb{E}_{\mathcal{D}_N,\gamma}[D_\ell(\bar{p}^*(y|x), \hat{p}_{N,\gamma}(y|\boldsymbol{x}))]}_{\text{generalized variance}}. \tag{7}$$

The first term in Eqn. 7 represents the intrinsic noise of the data-generating process, and therefore corresponds to

aleatoric uncertainty, which is irreducible with more training data. The second term captures the generalised bias, defined as the divergence between the mean predicted distribution $\bar{p}^*(y|\boldsymbol{x})$ averaged over data and procedural uncertainty, and the true conditional distribution $p(y|\boldsymbol{x})$. Finally, the third term reflects the generalized variance of the predictive distribution under data and procedural uncertainty. In Section 3.2, we will further decompose it into data and procedural components, by means of the conditional bias-variance decomposition for Bregman divergences (see (Gneiting & Raftery, 2007; Adlam et al., 2022)).

Connecting the bias component to Eqn. 5, one can decompose it into approximation error and estimation bias. Mainly the approximation error has seen some discussion in previous research on epistemic uncertainty, because the approximation error is nonzero when the ground-truth model is not in the hypothesis space, which might happen for certain hypothesis spaces, such as linear models. However, for universal function approximators, such as many types of deep neural networks, the approximation error will vanish (Hornik et al., 1989; Cybenko, 1989; Allenby & Labuschagne, 2009). Therefore, we believe that estimation bias deserves more discussion in modern deep learning research. It arises from the use of finite training datasets and from regularization procedures employed during training, which is inevitable to make learning possible with a finite dataset (Hastie et al., 2009). **Approximation error and estimation bias are both not included in existing epistemic uncertainty estimators, such as the one in Eqn. 3, supporting our position that epistemic uncertainty estimation is fundamentally incomplete**.

In the classical bias-variance decomposition for regression, the first term of Eqn. 7 is commonly estimated using Eqn. 3 as $\mathbb{E}[\hat{\sigma}^2]$, where $\hat{\sigma}^2$ denotes the predictive variance learned from the model. When minimizing squared loss, this quantity does not recover the true aleatoric noise (the details can be found in Appendix B.1). Instead, it satisfies $\mathbb{E}[\hat{\sigma}^2] = \mathbb{E}[(y - \hat{f}(x))^2] + \sigma^2$, as shown in (Zhang et al., 2024). Consequently, the learned aleatoric term absorbs observation noise and systematic prediction error: whenever $\hat{f}(x)$ deviates from the conditional mean, that deviation contributes to the learned $\hat{\sigma}^2$, inflating the aleatoric uncertainty estimation. Bias in the estimation of the mean will result in an overestimation of aleatoric noise.

### Running Example - Deep Ensembles

Figure 3 presents the uncertainty estimates produced by a Deep Ensemble on the synthetic dataset that we described earlier. The figure illustrates two key failure modes. First, in the top panel, the left side of the input domain exhibits a systematic prediction error: due to regularization, the model has not yet recovered the true

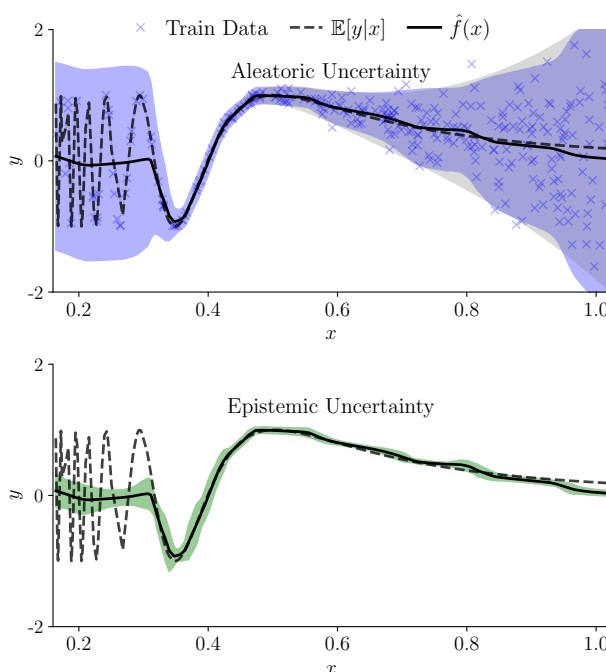

*Figure 3.* Deep Ensemble uncertainty estimates. The top row shows the training data points (blue crosses), the true conditional mean (dashed line), the model predictions (complete line), the 95% confidence interval for aleatoric uncertainty (purple), as well as the true 95% confidence interval of the aleatoric uncertainty (gray). The bottom row shows the estimate of epistemic uncertainty (green).

function in this region. Despite the observation noise being close to zero, the estimated aleatoric uncertainty is inflated, indicating that bias in the mean predictor is being absorbed into the aleatoric term. Second, in the bottom panel, the epistemic uncertainty remains low in the same region, even though the predictor is clearly misspecified with respect to the true function. **Taken together, these results show that, in the presence of systematic bias, deep ensembles can simultaneously overestimate aleatoric uncertainty and underestimate epistemic uncertainty, yielding an epistemic estimate that is inherently incomplete**.

Despite a complex function on the left side of the domain, and a high aleatoric uncertainty on the right side, the synthetic example that we consider is with a one-dimensional feature space very simple. However, if uncertainty estimation is already unreliable in such a simple setting, there is very few hope that things get better for modern high-dimensional datasets. In Appendix C.3 we observe exactly the same issues for a real-world regression dataset. Moreover, for other second-order methods, estimation bias also

leads to an overestimation of aleatoric uncertainty — see Appendix C.2 for additional experimental results that support this claim. In Bayesian methods, for instance, estimation bias is enforced by placing priors on the weights of the neural network. For simple hypotheis spaces, this allows for a flexible way of incorporating domain knowledge, but the choice of priors becomes quite arbitrary in large Bayesian neural networks (Wilson & Izmailov, 2020; Fortuin, 2022). Other methods, such as Monte Carlo Dropout and Bootstrap ensembles, also contain regularization techniques by design, which also lead to estimation bias.

### 3.2. Data and procedural components of epistemic uncertainty

In supervised learning, a trained predictor is a function of both the observed dataset and the learning procedure. A learning algorithm based on a neural network returns a predictive distribution $\hat{p}_{D_N,\gamma}(y|\boldsymbol{x})$, which is dependent on both the dataset $D_N$ as well as the procedural parameters $\gamma$. As a result, variance can be decomposed into a procedural component, formed by the expected variance induced by $\gamma$, while the data is fixed, and a data component, consisting of the variance induced by resampling datasets $D_N$. Using the law of total variance, for $\hat{y} \sim \hat{p}_{\mathcal{D}_N,\gamma}(y|\boldsymbol{x})$ we have

$$\begin{aligned} \text{Var}_{\mathcal{D}_N,\gamma}(\hat{y}|\boldsymbol{x}) = \ &\mathbb{E}_{\mathcal{D}_N}\left[\text{Var}_\gamma(\hat{y}|\boldsymbol{x})|\mathcal{D}_N\right] \\ &+ \text{Var}_{\mathcal{D}_N}\left(\mathbb{E}_\gamma[\hat{y}|\boldsymbol{x}]|\mathcal{D}_N\right). \end{aligned} \quad (8)$$

So far we claimed that second-order models ignore the bias component when estimating epistemic uncertainty. **Using Deep Ensembles as a running example, we also argue that common second-order methods only track parts of the epistemic uncertainty that originates from variance.**

---

**Running Example - Deep Ensembles**

**Deep Ensembles capture procedural uncertainty only.** Let $\gamma \sim P^\gamma$ denote the random seed (weight initialization, data ordering, etc.) that differentiates each ensemble member's training, where all members are trained on the *same* dataset $\mathcal{D}_N$. For the regression setting and a Deep Ensemble of size $d$, each member produces predictive parameters $\boldsymbol{\theta}_{\gamma_i}(\boldsymbol{x}) = (f_{\gamma_i}(\boldsymbol{x}), \sigma^2_{\gamma_i}(\boldsymbol{x}))$, for $i = 1, \ldots, d$. The empirical second-order distribution is a mixture of Dirac measures:

$$q(\boldsymbol{\theta}|x) = \frac{1}{d}\sum_{i=1}^{d}\delta_{\theta_{\gamma_i}(x)}(\theta).$$

Applying the variance-based decomposition (Eqn. 3) and taking $d \to \infty$, the epistemic estimate is:

$$\text{Var}_\gamma\left(\mathbb{E}_{y|\boldsymbol{x}}[\hat{y}|\boldsymbol{x},\gamma]\right) = \mathbb{E}_\gamma\left[(f_\gamma(\boldsymbol{x}) - \bar{f}(\boldsymbol{x}))^2\right],$$

---

where $\bar{f}(\boldsymbol{x}) = \mathbb{E}_\gamma[f_\gamma(\boldsymbol{x})]$. This term is linked to the first term in Eqn. 8, reflecting the procedural uncertainty. The data uncertainty component $\text{Var}_{D_N}(\mathbb{E}_\gamma[\hat{y}|\boldsymbol{x}]|\mathcal{D}_N)$, which captures how predictions would vary under different training datasets from the same distribution, is entirely absent from the Deep Ensemble's epistemic estimate.

While Deep Ensembles might explore multiple (local) optima of the training loss by using different random seeds (Wilson & Izmailov, 2020), Bayesian approaches attempt to explicitly approximate the posterior over parameters, which can yield more principled estimates of epistemic uncertainty. However, the practical methods used to approximate the posterior distribution also face restrictions: Approximate inference methods like the Laplace approximation or variational inference often capture only a *single* local mode (Wilson & Izmailov, 2020), employ simplifying assumptions (Psaros et al., 2023), or can be computationally costly in high-dimensional weight spaces. Other issues arise with methods that try to estimate the second-order distribution directly via loss minimisation, such as Deep Evidential Regression (Amini et al., 2020).

The theoretical analysis above raises an immediate question: Given a second-order uncertainty estimation method, how can one assess which components of epistemic uncertainty it represents? In the absence of observable ground-truth uncertainty, epistemic uncertainty is typically evaluated through downstream tasks (Gawlikowski et al., 2023) such as out-of-distribution detection (Hendrycks & Gimpel, 2017; Ovadia et al., 2019; Kopetzki et al., 2021), or selective prediction (Geifman & El-Yaniv, 2017). However, these benchmarks evaluate the *utility* of an uncertainty score for a particular decision problem, but they do not identify which component of uncertainty has been captured. We return to this point of view in Section 4.2. Here, **we argue that separately quantifying procedural and data uncertainty is only possible through statistical simulation**, where repeated sampling from the data-generating process can be performed or approximated. Following Jürgens et al. (2024), we formalize this through the notion of a *reference distribution*.

**Definition 3.1.** Consider a hypothesis space of models with $\phi \in \Phi$ model parameters and $\gamma \in \Gamma$ procedural parameters. We define the *reference distribution* w.r.t. $P^N$ and $P^\gamma$ as

$$\begin{aligned} q_{N,\gamma}(\boldsymbol{\theta}|\boldsymbol{x},\phi) &:= \mathbb{P}(\hat{\boldsymbol{\theta}}_{N,\gamma}(x) = \boldsymbol{\theta}) \\ &= \int_\Gamma \int_{\mathcal{X}\times\mathcal{Y}} \mathbb{I}\left[(\hat{\boldsymbol{\theta}}_{N,\gamma}(x) = \boldsymbol{\theta})\right]dP^N dP^\gamma, \end{aligned}$$

where $\hat{\boldsymbol{\theta}}_{N,\gamma}$ is the empirical estimate of $\boldsymbol{\theta}$, obtained via empirical risk minimization.

Intuitively, the reference distribution captures the full es-

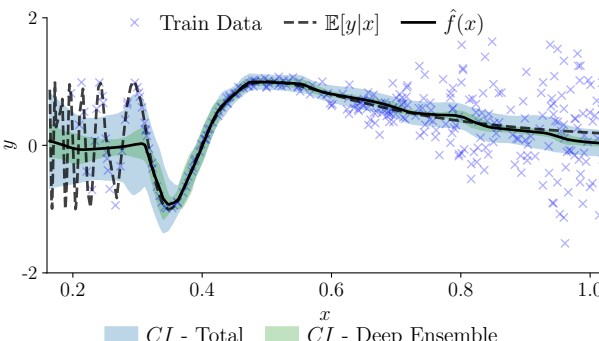

*Figure 4.* Comparison of reference distribution and Deep ensemble estimates. The confidence intervals of procedural and data variance as estimated by the reference distribution are shown in blue, while the Deep Ensemble estimate of epistemic uncertainty are shown in green.

timation uncertainty by marginalizing over both data uncertainty (variation across datasets $\mathcal{D}_N \sim P^N$) and procedural uncertainty (variation across $\gamma \sim P^\gamma$). Its variance decomposes exactly according to Eqn. 8, enabling direct comparison with the epistemic estimates produced by any second-order method. In practice, the reference distribution is estimated by training models on $n_d$ independent datasets with $n_\gamma$ random initializations each, yielding $n_d \times n_\gamma$ models whose empirical variance approximates the theoretical quantity. For real-world datasets where sampling from the data-generating distribution is infeasible, disjoint partitioning provides a practical approximation (see Appendix C.1).

> **Running Example - Deep Ensembles**
>
> We evaluate a Deep Ensemble of 10 members on the synthetic data example described in Eqn. 6. The reference distribution is estimated using $n_d = 20$ resampling steps and $n_\gamma = 10$ separate models are trained. The full experimental details are provided in Appendix C.1. Figure 4 illustrates that Deep Ensembles capture only procedural uncertainty, underestimating the variance from the reference distribution.

Further results on synthetic and real-world data for additional second-order models (Appendix C.2-C.3) reinforce our arguments: Figure 9 demonstrates that across all evaluated methods, epistemic uncertainty estimates fall systematically below the estimate of the reference distribution. Notably, Bayesian methods with richer posteriors produce higher epistemic estimates in low data regimes. However, this rather reflects their failure to learn the underlying data-generating distribution than the correct variance.

As bootstrapped ensembles are trained on resampled versions of $\mathcal{D}_N$, one might expect them to better capture the data uncertainty $\mathrm{Var}_{\mathcal{D}_N}(\mathbb{E}_\gamma[\hat{y}|\boldsymbol{x}]|\mathcal{D}_N)$, since under regular-

ity conditions, the bootstrap variance consistently approximates the true sampling variance (Efron, 1992). Recent work (Jain & Bates, 2025) decomposes bootstrap-based epistemic uncertainty estimates into data and procedural components. However, bootstrap-based estimates are inherently limited by the empirical support of the dataset and the estimation is only approximate in the finite sample regime. In our experiments (Appendix C.2-C.3) we find that the total epistemic uncertainty as given by the reference distribution is still underestimated.

## 4. Alternative Views

Our position is not that existing uncertainty methods are not useful at all, but that claims of estimating epistemic uncertainty must specify which uncertainty source is being represented. We discuss four alternative views and explain where they agree and disagree with our position.

### 4.1. Standard second-order methods sufficiently capture epistemic uncertainty

A common view is that second-order distributions already provide the right object for uncertainty disentanglement, and that no finer taxonomy is needed. Under this view, a method produces a second-order distribution over predictive distributions, from which aleatoric and epistemic components are extracted via decomposition formulas (Hüllermeier & Waegeman, 2021; Gawlikowski et al., 2023).

Many existing methods instantiate this view. Bayesian neural networks (Neal, 2012) place a prior over the network's weights and use the posterior predictive distribution to induce a second-order distribution. Due to the intractability of the posterior distribution induced by this hypothesis space (Kendall & Gal, 2017), one typically samples from it, using different approximation techniques, such as **Variational inference** (Jordan et al., 1999; Blundell et al., 2015; Blei et al., 2017), **Monte Carlo Dropout** (Gal & Ghahramani, 2016), **Laplace approximations** (MacKay, 1992; Daxberger et al., 2021), **Stochastic Weight Averaging Gaussian** (Maddox et al., 2019) or related sampling schemes. Ensemble methods construct an empirical second-order distribution by aggregating several trained predictors: **Deep Ensembles** (Lakshminarayanan et al., 2017) vary optimization randomness such as initialization and data ordering, whereas bootstrap ensembles additionally vary resampled versions of the training data. Gaussian-process-based and distance-aware variants, such as **Spectral-Normalized Gaussian Processes** (Liu et al., 2020a), induce uncertainty through structured output-space or feature-space distributions. In **Evidential Deep Learning** (Ulmer et al., 2023; Sensoy et al., 2018; Malinin & Gales, 2018), the parameters of the second-order distribution are learned directly, for example using Dirichlet distributions in classification (Sensoy et al., 2018) or

Normal-Inverse-Gamma distributions in regression (Amini et al., 2020).

We agree that this second-order perspective is useful and unifies a large part of the uncertainty quantification literature. Our disagreement is with the stronger interpretation that the variance given by the respective second-order distribution automatically entails all epistemic uncertainty of the underlying model. The second-order distribution rather represents the randomness induced by the specific construction, and is hence a method-dependent proxy. It generally does not include approximation error or estimation bias, and it estimates variance only partially. **Our position is not that second-order methods are irrelevant, but that their interpretation is underspecified.** They estimate a specific variance component represented by their own construction, not necessarily the full reducible uncertainty described in Sections 3.1 and 3.2. Claims that a method estimates epistemic uncertainty should therefore specify which components are represented, and which components are absent.

### 4.2. Downstream task performance is enough to validate uncertainty estimates

Another pragmatic view is that uncertainty estimates should be judged only by their usefulness in downstream tasks such as **out-of-distribution detection** (Hendrycks & Gimpel, 2017; Ovadia et al., 2019; Kopetzki et al., 2021), **robustness to adversarial attacks** (Malinin & Gales, 2019; Smith & Gal, 2018), **selective prediction** (Geifman & El-Yaniv, 2017) or **active learning** (Nguyen et al., 2022). According to this view, a decomposition does not need to be semantically faithful, as long as it improves the target decision problem. We agree that downstream performance is essential for deployment. However, downstream tasks test the utility of an uncertainty score for a particular decision rule, not whether the score faithfully represents epistemic uncertainty. In particular, downstream metrics usually do not distinguish between the different sources of epistemic uncertainty we defined in Section 2, and aleatoric uncertainty. A method can therefore perform well on a downstream task while estimating a different uncertainty object from the one claimed.

For out-of-distribution detection, the target is primarily distributional uncertainty, i.e. whether a test input is generated from a different distribution than the training data. Performance on this task can therefore be driven by the distributional shift, feature-space distance, support mismatch, or semantic mismatch, none of which implies that the method captures all parts of epistemic uncertainty. More fundamentally, Li et al. (2025) argue that out-of-distribution detection methods may be generally ill posed: uncertainty-based methods conflate high uncertainty with being far from training data, while feature-based methods conflate feature-space

distance with it. Selective prediction, on the other hand, has a different objective. A selective predictor abstains when prediction risk is high, and it has been shown that this risk is minimized by rejecting points with high total uncertainty (Hofman et al., 2026). It therefore does not require a faithful decomposition into aleatoric and epistemic components. Adversarial robustness benchmarks evaluate local sensitivity under a chosen perturbation model. The resulting failures can be caused by the geometry of the learned decision boundary, feature non-robustness, model bias, lack of training support, or distribution shift induced by the perturbation set. Testing adversarial robustness is hence a useful stress test for reliability, but not a direct test of uncertainty disentanglement. Finally, active learning is the downstream task most closely connected to epistemic uncertainty, because its goal is to select observations whose labels are expected to improve the learned predictor. Even here, however, task performance may reward heuristics that correlate with informativeness without isolating different parts of epistemic uncertainty.

**Downstream task benchmarks should therefore be interpreted as extrinsic utility tests, not as evidence that a method faithfully estimates epistemic uncertainty.** This motivates the usage of component-level diagnostics, using simulation tools such as the reference distribution described in Section 3.2. This way, one can explicitly analyse which part the different components of variation play, and compare it to the estimate of a specific method.

### 4.3. Aleatoric/epistemic uncertainty disentanglement is inherently flawed

A strong alternative view is that the distinction between aleatoric and epistemic uncertainty should be questioned altogether rather than refined. Recent work has questioned both the standard measures used for this decomposition and the conceptual adequacy of the binary taxonomy itself. Wimmer et al. (2023) show that conditional entropy and mutual information can violate desiderata for faithful uncertainty attribution, and additive decompositions of total uncertainty can be difficult to interpret. More broadly, Smith et al. (2025) argue that the terms aleatoric and epistemic are often used for distinct quantities, namely irreducible label noise, posterior dispersion, model misspecification, finite-sample variability, distribution shift, and value of information. Mucsányi et al. (2024) empirically show that no second-order method provides pairs of disentangled uncertainty estimates in practice.

We agree with the core diagnosis that the binary decomposition is too coarse. However, we draw a different conclusion. The problem is not that reducible uncertainty has no meaningful structure, but that many different reducible parts are collapsed into the single word *epistemic*. The parts of epis-

temic uncertainty correspond to different interventions such as changing the hypothesis space, adapting or stabilizing the learning procedure, collecting more training data, or detecting distribution shift. Epistemic uncertainty remains useful as an umbrella term for reducible uncertainty, but claims of estimating it are underspecified unless they state which reducible source is being estimated.

### 4.4. Alternative uncertainty representations avoid the problem

Given the documented limitations of second-order distributions, several alternative frameworks have emerged. We focus on deterministic methods based on distances or direct loss prediction, which have gained traction for computational efficiency and competitive empirical performance, then briefly discuss credal sets and conformal prediction.

**Deterministic methods.** Deterministic uncertainty quantification methods (Postels et al., 2022) quantify epistemic uncertainty using either the distribution of latent representations learned by the model, or by directly predicting the loss of the network's predictions. **Deterministic Uncertainty Quantification** (Van Amersfoort et al., 2020) replaces the softmax layer with an RBF-like output layer around class centroids, thereby measuring distance to the centroids as a proxy for uncertainty. The **Mahalanobis method** (Lee et al., 2018) captures epistemic uncertainty post-hoc by measuring the distance from the learned class-conditional distributions in feature space. **Direct Epistemic Uncertainty Prediction** (Lahlou et al., 2023) quantifies epistemic uncertainty as the excess risk of the predictor, i.e. the gap between pointwise expected loss and the loss of the Bayes predictor. It is learned via a secondary model trained on out-of-sample errors. Distance-based methods implicitly capture a form of distributional uncertainty: distance from training data correlates with regions where predictions are unreliable. However, **they do not explicitly decompose epistemic uncertainty into data and procedural components, nor do they account for systematic bias**. On the other hand, methods based on predicting the loss of the model are robust to model bias, but require training and maintaining an additional error-prediction model.

**Credal sets and imprecise probabilities.** Rather than placing distributions over distributions, credal sets represent epistemic uncertainty as convex sets of probability distributions (Walley, 1991). Recent work has operationalized credal sets in deep learning (Caprio et al., 2024b;a). Theoretically, credal sets offer a more expressive uncertainty language, however, they are not yet fully established in mainstream machine learning practice. Computational challenges arise from working with sets rather than point estimates, and the connection to standard training procedures remains less direct than for probabilistic methods.

**Conformal prediction.** Conformal prediction (Vovk et al., 2005; Angelopoulos et al., 2023) produces prediction sets with finite-sample coverage guarantees, which hold for any sample size and any underlying model. However, it addresses a different problem than epistemic uncertainty quantification, since it does not decompose uncertainty into aleatoric and epistemic components. Conformal methods are thus complementary to, rather than a replacement for, principled epistemic uncertainty estimation.

## 5. Conclusion and call to action

Epistemic uncertainty is generally not well defined and is still a heuristic concept for machine learning models, leading to interpretability issues for uncertainty estimates. Our analysis reveals that the prominent second-order uncertainty estimation methods systematically fail to capture what they claim to measure. Two mechanisms drive this failure: First, the predictive bias contaminates the aleatoric-epistemic uncertainty decomposition and should be evaluated as an epistemic component. Second, epistemic uncertainty estimates typically only reflect parts of the true variance, often capturing procedural rather than data uncertainty.

**Call to action.** We advocate for a more fine-grained taxonomy that explicitly distinguishes between model uncertainty, comprising approximation error plus estimation bias, and estimation uncertainty, comprising data and procedural components. New methods should be evaluated not only on downstream tasks but also against reference distributions that decompose variance into different sources. Claims of epistemic uncertainty estimation should specify which components are captured and acknowledge which are not. In safety-critical applications, we recommend: (1) assessing model bias separately through held-out validation or domain expertise, (2) treating epistemic uncertainty estimates as lower bounds on reducible uncertainty rather than complete characterizations, and (3) using multiple complementary uncertainty methods, as different methods capture different uncertainty components.

## Acknowledgments

S.J. was supported by the Research Foundation Flanders (FWO Vlaanderen) (PhD fellowship, fundamental research grant 11A3G26N). M.J. and W.W. received funding from the Flemish government under the "Flanders AI Research" program.

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

## A. Impact of recent literature

In recent years, uncertainty disentanglement has gained importance in the machine learning literature. Considering the following collection of highly relevant papers, they have received almost 40,000 citations according to Google Scholar: (Der Kiureghian & Ditlevsen, 2009; Kendall & Gal, 2017; Hüllermeier & Waegeman, 2021; MacKay, 1992; Depeweg et al., 2018; Malinin & Gales, 2018; 2019; Amini et al., 2020; Lakshminarayanan et al., 2017; Wilson & Izmailov, 2020; Gal et al., 2016; Charpentier et al., 2020; 2022; Stadler et al., 2021; Lee et al., 2018; Liu et al., 2020b; Van Amersfoort et al., 2020; Sun et al., 2022; Deng et al., 2023)

## B. Incomplete uncertainty estimators

### B.1. Bias aleatoric uncertainty estimators

For a data-generating process such that $y_i = f(\boldsymbol{x}_i) + \epsilon_i$, where $\epsilon_i \sim \mathcal{N}(0, \sigma^2(\boldsymbol{x}_i))$ is data-dependent (*heteroscedastic*) noise, and $f$ is a deterministic function, such that $p(\cdot|\boldsymbol{x})$ forms the density of a Gaussian distribution with mean $f(\boldsymbol{x})$ and variance $\sigma^2(\boldsymbol{x})$. There are multiple ways to model aleatoric uncertainty. One popular way in the literature to do so (Zhang et al., 2024) is using the variance attenuation (Lakshminarayanan et al., 2017; Amini et al., 2020; Valdenegro-Toro & Mori, 2022), which means to train a neural network with two heads (one that outputs the predictive mean and the other that outputs the predictive variance) using a loss function that reflects the likelihood of the observed data:

$$\ell_{va} = \frac{1}{M} \sum_{i=1}^{N} \frac{(y_i - \hat{f}(x_i))^2}{2\hat{\sigma}^2(x_i)} + \frac{\log(\hat{\sigma}^2(x_i))}{2}. \tag{9}$$

We rely on the derivation presented by (Zhang et al., 2024) to show how Eqn. 9 does not produce an unbiased estimator of the aleatoric uncertainty $\sigma^2(\boldsymbol{x})$, and refer the reader to its paper for all the details. In this setting, the expected value of the predictive variance becomes

$$\mathbb{E}[\hat{\sigma}^2(x)] = (f(x) - \hat{f}(x))^2 + \sigma^2(x). \tag{10}$$

Where one can see that the expected aleatoric noise differs from the true aleatoric noise by a square difference of the predictive mean and the true label.

While the original contribution of (Nix & Weigend, 1994) suggest to perform a warm-up period, one can see that this modification to the training process would improve the mean prediction, by fitting the model in a simpler loss landscape, and therefore the accuracy of $\hat{f}$, leaving the expected value of the predictive variance unchanged. Other methods have been proposed to produce a better estimate of aleatoric uncertainty. (Seitzer et al., 2022) proposed adding an extra term $\lfloor \hat{\sigma}^{2\beta}(x) \rfloor$ to the second term of $\ell_{va}$, where $\lfloor \cdot \rfloor$ denotes the stop gradient operation. However, this modification does not change the expected value of the estimator, it just restricts the learning path of the predictive mean. (Sluijterman et al., 2025) proposed a method that divides the training process into two parts, by creating some modified datapoints for the training of the second part. This process improves mean predictions and the potential to avoid overfitting, while the bias of the expected aleatoric uncertainty estimation is still there when the predictive mean deviates from the true value.

### B.2. Regularization effects on predictive variance

Neural networks are trained with multiple sources of regularization due to the limited sample size (e.g. early stopping, weight decay, dropout). A body of works in the literature, from textbooks to recent machine learning papers (Larsen & Hansen, 1994; Nusrat & Jang, 2018; Ninness & Hjalmarsson, 2004; Girosi et al., 1995), show how regularization reduces predictive variance by constraining how strongly the learned model can respond to finite-sample fluctuations in the training data, which happens through parameter shrinkage, effective dimensionality reduction, and smoothing of the learned function.

By reducing the variance, the third term of Eqn. 7 shrinks and the second term of the same equation increases, as bias increases from regularization. Our argument is key in this aspect: if we aim to capture epistemic uncertainty, looking only at the variance, which has been affected by the training process, gives an incomplete picture of the whole epistemic uncertainty.

## C. Experimental details

The code to reproduce all the experiments in this paper is available on this GitHub repository.

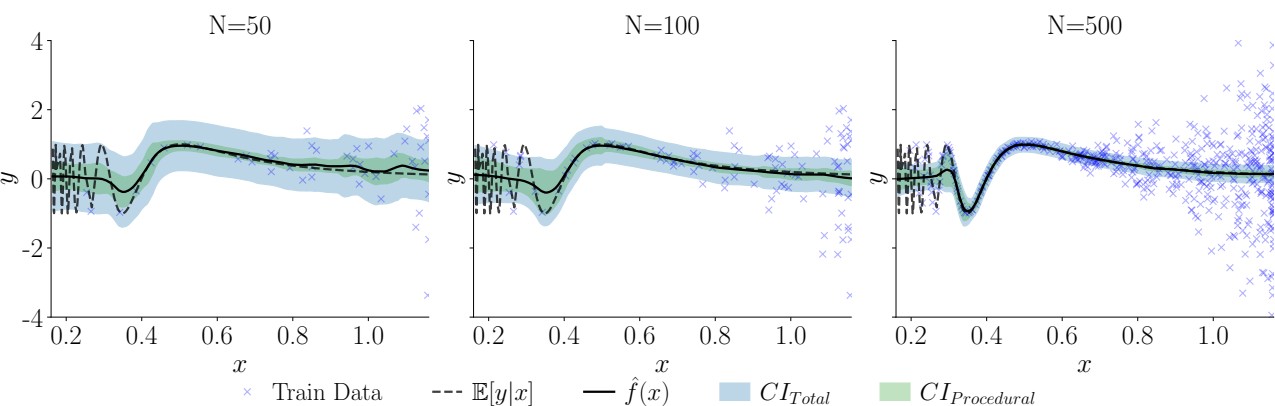

*Figure 5.* Estimated procedural and data uncertainty of the reference distribution for increasing sample sizes. The reference distribution is estimated by resampling training data and procedural parameters, using $n_\gamma = 10$ and $n_d = 20$. Procedural and data uncertainty are calculated using the decomposition formula in Eqn. 7. Even when the model predictions (complete line) have an important deviation (bias) from the ground-truth expected value (dashed line), the epistemic uncertainty estimate decreases with more data.

### C.1. Reference distribution approximation

**Synthetic dataset.** To obtain an empirical approximation of the reference distribution (see Definition 3.1), we sample 20 different datasets from the known data-generating process to approximate the data uncertainty component of the epistemic uncertainty. For each dataset, we train 10 different MLP models with different random weight initializations, which allows us to obtain a procedural uncertainty estimate. Each MLP uses the whole dataset. The MLP models have 4 hidden layers with 100 neurons each. We use a learning rate of 0.009 and train the models for 500 epochs. We repeated the process for different training dataset sizes: $50, 100, 500$ with batch sizes $32, 32, 64$ respectively. Seeds for the datasets are $\{7, 42, 2024, 123, 999, 50, 100, 150, 200, 250, 300, 350, 400, 450, 500, 550, 600, 650, 700, 750\}$.

Figure 5 shows the obtained procedural and data uncertainty of the reference distribution for different training dataset sizes. First, we see that the total variance of the reference distribution decreases when sample size increases, supporting the claim that the reference distribution properly represents data uncertainty. Second, we see that the estimate of the procedural uncertainty also decreases with an increasing sample size, which is not unexpected, because initial procedural configurations become less important when more data is observed. Third, in the region with high epistemic uncertainty, the reference distribution has a very large bias, due to the underlying complexity of the data-generating function. The bias is in fact, so high that aleatoric uncertainty cannot be correctly estimated anymore. This last finding confirms our central position: Any claim to properly represent epistemic uncertainty must examine not only how different sources of uncertainty are modeled, but also whether bias is systematically distorting the partitioning of uncertainty. When the bias is high, estimates of both epistemic and aleatoric uncertainty can be completely misleading.

**Real dataset.** To estimate the reference distribution as described in Section 3.2, we partition the training dataset into 100 disjoint subsets of $12,067$ observations each (i.e., $n_d = 100$). This partition allows us to approximate the data uncertainty component from the epistemic uncertainty. We argue that in big datasets, where enough disjoint subsamples can be taken, it is possible to approximate the variance coming from different datasets at training time. For each subset, we train $n_\gamma = 5$ multilayer perceptrons (MLPs) with different random weight initializations, this part of the process is the same as in the synthetic data setting and results in an approximation of the procedural component of the epistemic uncertainty. This whole process yields a total of 500 independently trained models. Each model produces predictive means $\hat{y}_{N,\gamma}(\boldsymbol{x})$, and by computing the empirical variance across these 500 predictions, we obtain an estimate of $\text{Var}_{\mathcal{D}_N,\gamma}(\hat{y}|\boldsymbol{x})$ as defined in Eqn. 8. This procedure operationalizes the reference distribution introduced in Definition 3.1: the pseudo-resampling across datasets ($\mathcal{D}_N$) captures data uncertainty while the random initializations account for procedural uncertainty. Given the large sample size of the dataset, the partitioning approach provides a computationally feasible approximation of repeated sampling from the underlying data-generating process, effectively *emulating* access to its stochastic variability. This design allows us to decompose the epistemic uncertainty component into its data and procedural parts in a controlled manner, while being in a real-world data setup.

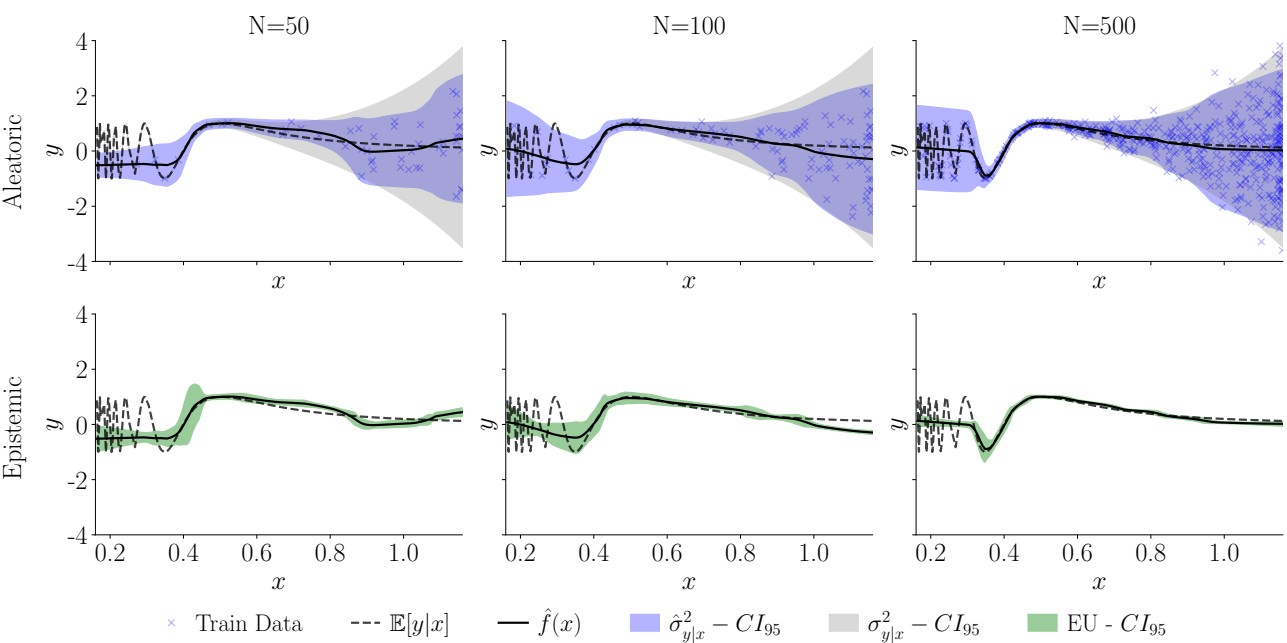

*Figure 6.* Deep Ensemble estimates, obtained with different sample sizes $N \in \{50, 100, 500\}$. The top row shows the training data points (blue crosses), the true conditional mean (dashed line), the model predictions (complete line), the 95% confidence interval for aleatoric uncertainty (purple), as well as the true 95% confidence interval of the aleatoric uncertainty (gray). The bottom row shows the estimate of epistemic uncertainty (green).

## C.2. Synthetic data experiments - Regression

To illustrate the limitations of existing uncertainty quantification methods in producing reliable disentangled estimates, we showcase the behavior of eight representative methods of the three second-order quantification approaches we discuss in the main paper. This section shows the experimental details and results from the uncertainty quantification methods applied to the problem described by Eqn. 6.

### C.2.1. EXPERIMENTAL ILLUSTRATIONS

We use the variance-based split of the predictive variance as in Eqn. 8 to evaluate the methods in the heteroscedastic regression setting. Figure 6 illustrates that epistemic uncertainty estimates decrease monotonically with sample size, consistent with the behavior observed for the reference distribution. This trend confirms that, as the dataset grows, models become increasingly confident, yet not necessarily more accurate, particularly in difficult-to-learn regions. In these settings, even with abundant data, the model can remain overconfident in systematically biased predictions. Such behavior reveals a key limitation of current epistemic uncertainty measures and disentanglement frameworks: they fail to account for systematic bias, leading to an underestimation of epistemic uncertainty. This pattern is not unique to Deep Ensembles, but is consistently observed across other methods (see Figures 12–18), indicating that this limitation is intrinsic to the way epistemic uncertainty is commonly represented rather than a method-specific artifact. Bias is approximated as specified in Eqn. 7: datasets are repeatedly resampled from the true data-generating process, models are trained on each sample, and the mean deviation of the predictions from the true conditional mean is computed. Table 1 shows that bias is frequently larger than –and largely uncorrelated with– the corresponding epistemic uncertainty estimates. These findings underscore our central position: any attempt to disentangle aleatoric from epistemic uncertainty must assess not only how each component is modeled, but also the extent to which systematic bias distorts their partitioning.

Figures 7c and 9 further illustrate the relationship between the epistemic uncertainty estimates and the procedural and data uncertainty components obtained from the reference distribution. When comparing these components, the relation shifts upward once data uncertainty is included alongside procedural uncertainty, indicating that most methods primarily capture the procedural component of epistemic uncertainty. This observation highlights the necessity of adopting a more fine-grained taxonomy to identify which parts of epistemic uncertainty are being represented, as well as the importance of

**Training sample size:** $N = 100$

| Model | $\hat{\sigma}(x)$ | $\mathrm{Var}\big[\hat{y}|\boldsymbol{x}\big]$ | $\mathbb{E}_{\mathcal{D}_N}[\hat{y}(x) - f(x_i)]$ | $\hat{\sigma}(x) - \sigma(x)$ |
|---|---|---|---|---|
| Reference | 0.83 (0.01) | 0.041 (0.005) | 0.084 (0.002) | 0.248 (0.004) |
| Bootstrap | 0.87 (0.167) | 0.006 (0.008) | 0.125 (0.022) | 0.239 (0.027) |
| DeepEnsemble | 0.87 (0.148) | 0.007 (0.011) | 0.147 (0.018) | 0.235 (0.021) |
| EDL | 0.21 (0.002) | 0.003 (0.0) | 0.269 (0.01) | 0.751 (0.005) |
| ViELBONet | 0.95 (0.064) | 0.079 (0.001) | 0.127 (0.021) | 0.234 (0.022) |
| MCDropoutNet | 0.90 (0.097) | 0.009 (0.002) | 0.190 (0.026) | 0.334 (0.033) |
| HMCNet | 9.08 (13.734) | 2.131 (0.224) | 0.179 (0.015) | 7.870 (0.853) |
| LaplaceNet | 2.43 (5.559) | 4.211 (3.047) | 0.133 (0.024) | 0.800 (0.540) |
| GP | 0.86 (0.048) | 0.008 (0.001) | 0.194 (0.013) | 0.308 (0.020) |

**Training sample size:** $N = 500$

| Model | $\hat{\sigma}(x)$ | $\mathrm{Var}\big[\hat{y}|\boldsymbol{x}\big]$ | $\mathbb{E}_{\mathcal{D}_N}[\hat{y}(x) - f(x_i)]$ | $\hat{\sigma}(x) - \sigma(x)$ |
|---|---|---|---|---|
| Reference | 1.04 (0.016) | 0.013 (0.001) | 0.044 (0.001) | 0.076 (0.003) |
| Bootstrap | 0.98 (0.100) | 0.003 (0.004) | 0.085 (0.001) | 0.131 (0.010) |
| DeepEnsemble | 0.99 (0.136) | 0.004 (0.005) | 0.100 (0.002) | 0.122 (0.010) |
| EDL | 0.67 (0.247) | 0.013 (0.007) | 0.168 (0.016) | 0.967 (0.031) |
| ViELBONet | 1.06 (0.115) | 0.009 (0.000) | 0.097 (0.005) | 0.122 (0.012) |
| MCDropoutNet | 1.06 (0.079) | 0.008 (0.003) | 0.119 (0.006) | 0.163 (0.008) |
| HMCNet | 4.17 (20.356) | 0.378 (0.017) | 0.129 (0.003) | 1.936 (1.495) |
| LaplaceNet | 1.29 (3.777) | 0.487 (3.802) | 0.115 (0.016) | 0.187 (0.345) |
| GP | 1.04 (0.069) | 0.008 (0.003) | 0.172 (0.005) | 0.152 (0.014) |

*Table 1.* Aleatoric and epistemic uncertainty estimates, bias, and distance from the aleatoric uncertainty to ground truth noise across models. "Referece" refers to the variance and predictive mean as estimated by the reference distribution. Values are means over the test set from five runs; standard deviations are in parentheses.

the reference distribution in empirically validating these estimates. On the other hand, the aleatoric uncertainty estimates on the right-hand side of the data support improve as the sample size increases, the behaviour on the left-hand side remains problematic. Specifically, the model persistently overestimates aleatoric uncertainty. A plausible explanation is that the approximate symmetry of the observations around the learned conditional mean induces the model to misinterpret systematic deviation as inherent data noise. Consequently, this misestimation is not corrected with additional data; instead, the model becomes increasingly confident in its erroneous predictions and continues to interpret the discrepancy from the ground-truth conditional mean as aleatoric uncertainty. This behavior is illustrated in Figure 8a, which compares the average estimated aleatoric uncertainty with the true aleatoric noise in regions of low and high model bias. The figure shows that in areas of high bias, the estimated aleatoric uncertainty diverges substantially from the true value, demonstrating that biased predictions systematically distort the disentanglement of aleatoric and epistemic uncertainty.

The results for the other methods can be seen in Figures 12-18. We highlight two additional key observations: First, we observe that Bayesian approaches –specifically Laplace Approximation (LA) and Hamiltonian Monte Carlo (HMC)– require a larger amount of data to produce accurate estimates of aleatoric uncertainty in the right-hand region of the input space. This behavior aligns with theoretical expectations of Bayesian inference: The increased data requirement reflects the greater difficulty these models encounter in learning the underlying data-generating function, which in turn leads to higher epistemic uncertainty–consistent with models that make inaccurate predictions. Second, the Deep Evidential Regression (DER) method consistently produces poor estimates of aleatoric uncertainty across all sample sizes and throughout the entire input domain. These inaccuracies are accompanied by overconfident predictions, as evidenced by the systematic underestimation of epistemic uncertainty. This flawed characterization of uncertainty is consistent with findings reported in recent literature (Jürgens et al., 2024). We see that every method with a low epistemic uncertainty estimate overestimates the aleatoric uncertainty in the left-hand side of the figure due to their inability to acknowledge the bias as part of their epistemic uncertainty representations. Also, for more data-intensive methods, such as the Bayesian approaches, a more careful and realistic estimate of the epistemic uncertainty is obtained.

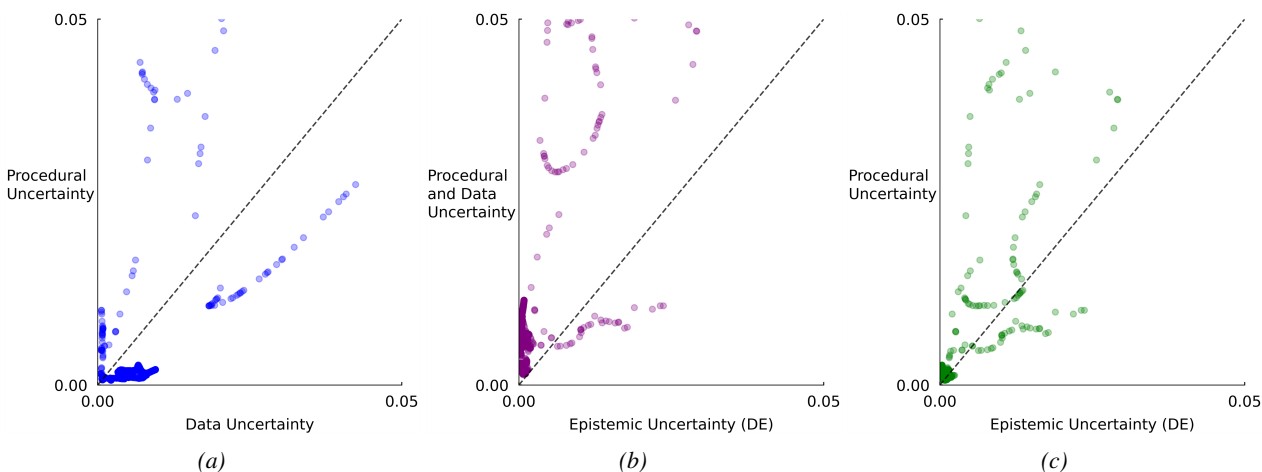

*Figure 7.* Epistemic uncertainty estimates for the synthetic dataset. (a)Decomposition of the epistemic uncertainty estimated with the reference distribution into procedural and data uncertainties. There exists a strong correlation between both, which is expected, and procedural uncertainty outweighs across almost all the data points. (b)Comparison between the epistemic uncertainty estimated with the reference distribution (both data and procedural components) and the epistemic uncertainty estimated with the Deep Ensemble. Here, the epistemic part of the reference distribution is larger, showing that the Deep Ensemble underestimates this uncertainty. (c)When the data uncertainty component is removed from the estimate of the reference distribution, both estimates are much closer, suggesting that the Deep Ensemble fails to capture the data uncertainty component of the epistemic uncertainty.

### C.2.2. IMPLEMENTATION DETAILS BY METHOD

All methods are evaluated for different training dataset sizes: $\{50, 100, 500\}$ with batch sizes $\{32, 32, 64\}$ respectively. The experiments are repeated for 5 different seeds: $7, 42, 123, 999$ and $2024$. To conduct the experiments, a hyperparameter tuning over relevant parameters was performed.

**Deep Ensemble.** Trained with an MLP of 4 hidden layers with 100 neurons each, learning rate of 0.009. Training epochs: $50, 250, 500$. Every member of the deep ensemble uses the whole dataset.

**Bootstrap Ensemble.** Trained as the Deep Ensemble, but each ensemble member uses a boostrap sample with $60\%$ of the training data.

**Monte Carlo Dropout.** A single MLP of 4 hidden layers with 100 neurons each is trained. A dropout rate of 0.2 is used at training and inference time. A learning rate of 0.002 is used.

**Variational Inference.** This model is trained based on Blundell et al. (2015). An MLP of 4 hidden layers with 100 neurons each converted to a Bayesian neural network is used. The model is trained with a learning rate of 0.005, 200 burn-in epochs, a beta value of 5, 10 Monte Carlo samples at training time, and 500 at test. The model is evaluated for 250 and 500 training epochs.

**Laplace Approximation.** An MLP of 4 hidden layers with 100 neurons each is used to find the MAP estimate, together with a learning rate of 0.005. For the Laplace step, a prior precision of 1 and a noise value of 2 are used. The model is trained for $200, 400$, and $800$ epochs, and uses 1000 samples from the posterior distribution. When using a training sample size of 500, a batch size of 128 is used.

**Hamiltonian Monte Carlo.** An MLP of 4 hidden layers with 100 neurons each is used as the underlying model. The sampling step consists of 200 samples with a sampling step of 0.00015. The first 50 samples are always burnt. A $\tau$ value of 1 is used as the indicator for a *bendy* function. At inference time, we sample 1000 times from the posterior distribution and drop the first 50. The model is trained for $1000, 2000$, and $3000$ epochs. When using a sample size of 500, a batch size of 128 is used.

**Deep Evidential Regression.** Leaning on the setup of Amini et al. (2020), an MLP of 4 hidden layers with 100 neurons each is used as the underlying model. The model is trained with a learning rate of 0.0003 over 5000 epochs, and a regularization strength of $\lambda = 0.01$.

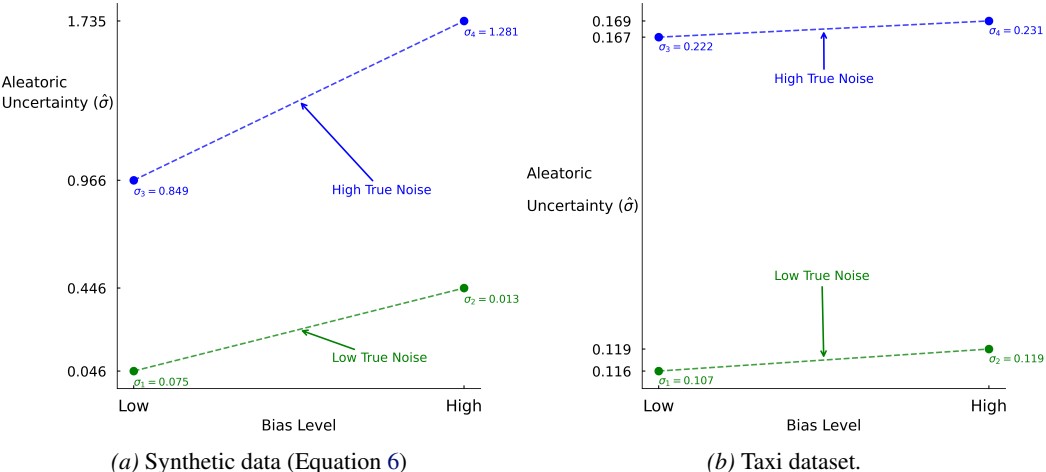

*(a)* Synthetic data (Equation 6)  *(b)* Taxi dataset.

*Figure 8.* Comparison of the estimated aleatoric uncertainty with the true (simulated) aleatoric noise across regions characterized by different levels of model bias and inherent data noise. The notation $\sigma_i$ for $i \in 1, 2, 3, 4$ denotes the true aleatoric uncertainty for each of the four regions: 1. low bias and low true noise, 2. high bias and low true noise, 3. low bias and high true noise, and 4. high bias and high true noise. For the synthetic dataset, the division between high- and low-noise regions follows Eqn. 6: points with $x < 0.6$ are considered low-noise, while those with $x \geq 0.6$ are considered high-noise. Within each noise region, samples are ranked according to their model bias, and the lower half of points (by bias magnitude) defines the low-bias region, while the upper half defines the high-bias region. For the taxi dataset, the high- and low-noise regions are determined using the 90-th percentile of the simulated true aleatoric uncertainty, as described in Section C.3. Within each noise region, data points are further divided into low- and high-bias subsets based on the 70th percentile of the estimated model bias. This construction allows for a direct comparison between the estimated and true aleatoric uncertainty across systematically defined regions, highlighting how model bias affects the reliability of aleatoric uncertainty estimates in both synthetic and real-world scenarios.

**Heteroscedastic Gaussian Process.** Leaning on the variational-sparse recipe of Titsias (2009), we use the GPytorch ApproximateGP module to train a two-output GP, trained with an RBF kernel. The number of inducing points (see (Hastie et al., 2009) for an explanation) is set to 256, the learning rate to 0.005, epochs to 2000.

### C.3. Real data experiments

In this section we extend our analysis to the New York taxi trip duration dataset published in this Kaggle competition. The dataset consists of taxi trip records from New York City, where the target variable corresponds to the trip duration and the feature space comprises 14 input variables. The training set contains $1,206,737$ observations, and the test set includes $230,406$ observations. This dataset is particularly suitable for evaluating uncertainty estimation methods for several reasons. First, its large sample size allows us to analyze how epistemic uncertainty behaves when the data-driven component becomes relatively small compared to the model and procedural components, allowing us to estimate data and procedural uncertainty via the reference distribution as we explained in Appendix C.1. Second, since the task is formulated as a heteroscedastic regression problem, where trip durations exhibit high variability depending on factors such as traffic conditions or route choice, it provides a natural setting for disentangling aleatoric from epistemic uncertainty. Finally, the dataset's scale and heterogeneity make it a realistic benchmark for assessing whether uncertainty quantification methods can remain well-calibrated and informative in complex, high-dimensional environments. In the same sense, a simulated estimate of ground truth aleatoric uncertainty can be obtained as we explained in Appendix C.3.1. As in the synthetic data experiments, we assume that the target variable follows a normal distribution, and the model outputs both the mean and variance of this distribution.

#### C.3.1. GROUND TRUTH ALEATORIC UNCERTAINTY APPROXIMATION

**Aleatoric uncertainty**. To approximate the ground-truth aleatoric uncertainty, we train an ensemble of $n_\gamma = 5$ multilayer perceptrons (MLPs) on the full training dataset of $1,206,737$ observations. Since aleatoric uncertainty represents the irreducible noise inherent to the data-generating process, training on the complete dataset minimizes the influence of data-driven and procedural components, yielding the most reliable estimate of this term attainable in practice. We take the mean predicted variance across the ensemble members as the reference estimate of aleatoric uncertainty. To assess the

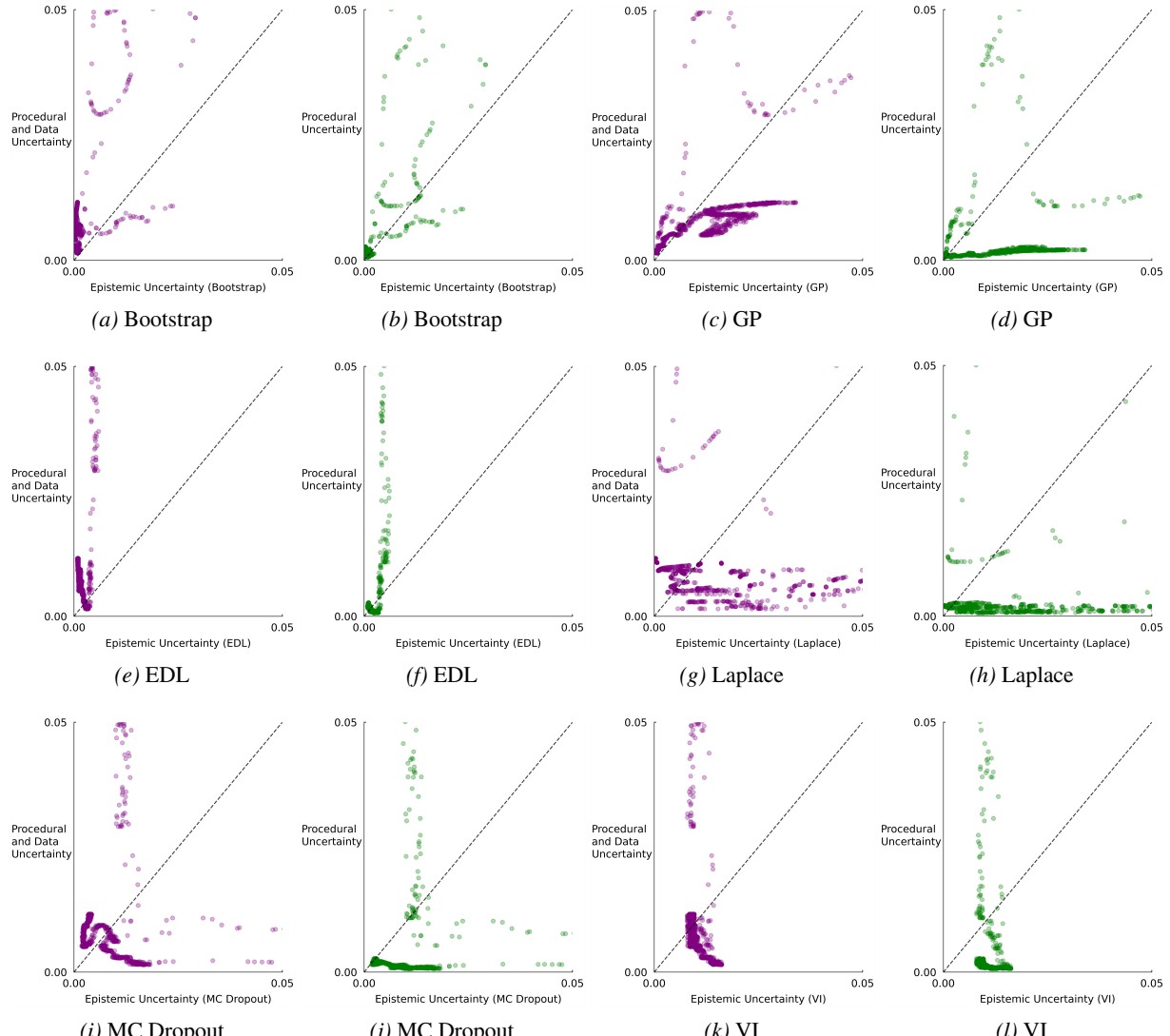

*Figure 9.* Comparison of procedural and data uncertainties vs epistemic uncertainty across methods on the synthetic regression dataset (Eqn. 6).

effect of limited data and model bias on uncertainty disentanglement, we then compare this reference estimate with those obtained when each model is trained using only $1\%$ of the available data. This comparison highlights how insufficient data coverage and higher epistemic uncertainty can distort the estimated aleatoric component, revealing the extent to which current methods conflate epistemic and aleatoric sources of uncertainty.

### C.3.2. EXPERIMENTAL ILLUSTRATIONS

Figure 10c shows the resulting decomposition of total epistemic variance into procedural and data components. We observe that procedural uncertainty dominates across most of the input domain. This outcome reflects the high capacity and non-convex optimization landscape of neural networks: even with abundant data, different random initializations can converge to distinct local minima, leading to substantial procedural variation. By contrast, data uncertainty is smaller, indicating that resampling subsets from such a large dataset induces only limited diversity in the fitted models. It is important to note that the relative magnitudes of these two components depend on the experimental configuration. Increasing the subset size $n_d$ would typically reduce data uncertainty, while enlarging $n_\gamma$ would shrink the procedural uncertainty. As discussed in Example 3.2 and in the synthetic data experiments, Deep Ensembles predominantly capture the procedural component of epistemic uncertainty. This limitation arises because ensemble diversity originates solely from random initialization

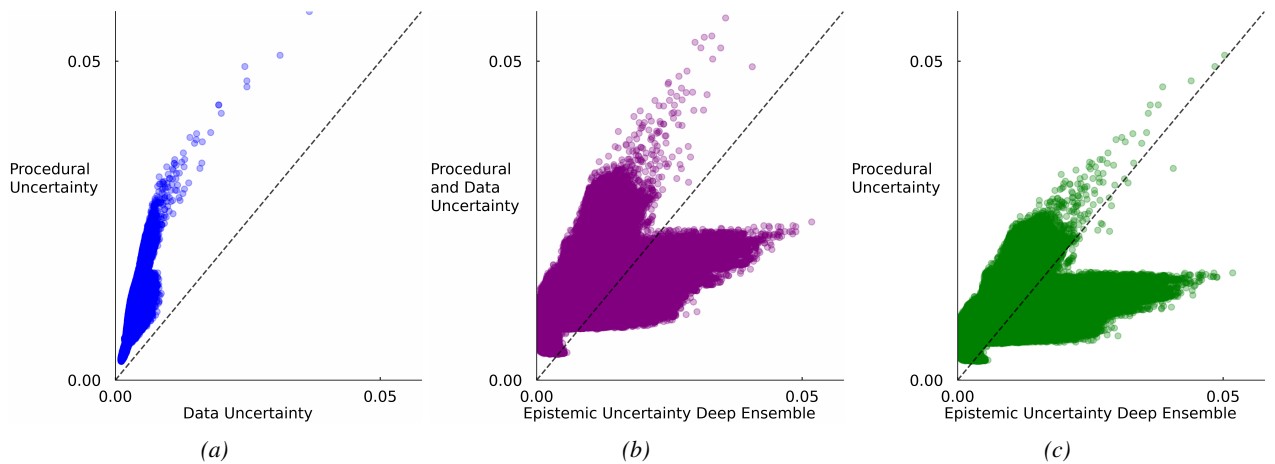

*Figure 10.* Epistemic uncertainty evaluation for the Deep Ensemble on the taxi dataset. (a) Decomposition of the variance of the reference distribution into data and procedural uncertainty. (b) Deep Ensemble epistemic estimates vs. the total epistemic uncertainty given by the reference distribution. 30% of points fall below the diagonal. (c) Deep Ensemble epistemic estimates vs. the procedural component of the reference distribution only.

and stochastic training dynamics, while all models are trained on the same dataset. Consequently, the variance across ensemble members reflects the sensitivity of the optimization process rather than uncertainty about the data distribution itself. This behavior is observed in Figure 10b, where the total variance of the reference distribution, represented on the $y$-axis as the sum of data and procedural uncertainty, is systematically higher than the epistemic uncertainty predicted by the Deep Ensemble. Quantitatively, the epistemic uncertainty estimate underestimates the reference distribution variance in approximately 80% of the predictive samples. This systematic underestimation indicates that the ensemble fails to account for the data-driven component of epistemic uncertainty, which captures how model predictions would vary under different training datasets drawn from the same data-generating process. To further examine this relationship, Figure 10a decomposes the reference distribution into its procedural and data uncertainty components. When comparing only the procedural component of the reference distribution with the epistemic uncertainty predicted by the Deep Ensemble, the discrepancy disappears. This alignment confirms that Deep Ensembles primarily quantify procedural uncertainty, that is, the uncertainty arising from stochasticity in optimization and model initialization, while largely neglecting the variability induced by finite data sampling. In practical terms, this result reinforces the theoretical argument presented in Section 3.2: the epistemic uncertainty estimated by Deep Ensembles does not reflect the full reducible uncertainty in the model but only its procedural subcomponent. Consequently, downstream tasks that rely on a correct epistemic uncertainty estimation, such as active learning, model selection, or out-of-distribution detection, may be misled by an incomplete uncertainty representation. In the same sense, Figure 8b shows that in the region where the model is more incorrect, the difference between the estimated aleatoric uncertainty and the simulated ground truth aleatoric uncertainty is larger than in the region where the model is more accurate, showing how a high bias can distort the disentanglement of aleatoric and epistemic uncertainty. The findings from this real-world regression task, therefore, underscore the importance of distinguishing between procedural and data uncertainty when evaluating or designing second-order uncertainty quantification methods.

| Model | $\hat{\sigma}^2$ | $\hat{\sigma}^2(x) - \sigma_*^2(x)$ | $\text{Var}[\hat{y}|\boldsymbol{x}]$ | $\text{Var}_{\mathcal{D}_N,\gamma}(\hat{y}|\boldsymbol{x}) - \text{Var}[\hat{y}|\boldsymbol{x}]$ | $\mathbb{E}_{\mathcal{D}_N}[\text{Var}_\gamma(\hat{y}|\boldsymbol{x})|\mathcal{D}_N] - \text{Var}[\hat{y}|\boldsymbol{x}]$ |
|---|---|---|---|---|---|
| DE | 0.1222 | 0.0004 | 0.0083 | 0.0046 | 0.0013 |
| Bootstrap | 0.1127 | 0.0091 | 0.0082 | 0.0047 | 0.0014 |
| MC Dropout | 0.1284 | 0.0066 | 0.0106 | 0.0023 | 0.001 |
| VI | 1.8859 | 1.7641 | 0.0055 | 0.0074 | 0.0041 |
| Laplace | 0.1754 | 0.0536 | 0.0137 | 0.0008 | 0.0041 |
| HMC | 0.2319 | 0.1101 | 0.1148 | 0.1019 | 0.1052 |
| GP | 1.0334 | 0.9116 | 0.0636 | 0.0507 | 0.054 |

*Table 2.* Taxi dataset uncertainties. $\text{Var}[\hat{y}|\boldsymbol{x}]$ represents the epistemic uncertainty estimated by each model. $\text{Var}_{\mathcal{D}_N,\gamma}(\hat{y}|\boldsymbol{x})$ corresponds to the epistemic uncertainty obtained with the reference distribution, while $\mathbb{E}_{\mathcal{D}_N}[\text{Var}_\gamma(\hat{y}|\boldsymbol{x})|\mathcal{D}_N]$ is the procedural uncertainty obtained by the reference distribution. The values are the average over all test data points.

### C.3.3. DATA PREPROCESSING

The raw dataset used for this task can be found in this Kaggle competition. The goal is to predict the duration of a taxi trip in New York. The dataset contains $1,458,644$ observations from 2016. The code to preprocess the data is in the GitHub repository of this paper. The target variable is transformed according to $y' = \log(y + 1)$.

**Data splitting**. We consider the data before June as the training dataset and the data after June as the test dataset. In this way, we get $1,206,737$ training points and $230,406$ test points. However, only $1\%$ of the dataset is used for training the models.

### C.3.4. IMPLEMENTATION DETAILS BY METHOD

**Deep Ensemble.** Trained with an MLP of 10 hidden layers with 300 neurons each, learning rate of 0.0009. Training epochs: 50. Every member of the deep ensemble uses the whole dataset (containing $1\%$ of the total training dataset).

**Bootstrap Ensemble.** Trained as the Deep Ensemble, but each ensemble member uses a boostrap sample with $60\%$ of the training data (out of the $1\%$).

**Monte Carlo Dropout.** A single MLP of 10 hidden layers with 300 neurons each is trained. A dropout rate of 0.2 is used at training and inference time. A learning rate of 0.0009 is used.

**Variational Inference.** This model is trained based on Blundell et al. (2015). An MLP of 10 hidden layers with 300 neurons each converted to a Bayesian neural network is used. The model is trained with a learning rate of 0.0009, 20 burn-in epochs, a beta value of 10, 10 Monte Carlo samples at training time, and 500 at test. The model is evaluated for 50 training epochs.

**Laplace Approximation.** An MLP of 4 hidden layers with 100 neurons each is used to find the MAP estimate, together with a learning rate of 0.0009. For the Laplace step, a prior precision of 1 and a noise value of 0.1 are used. The model is trained for 50 epochs, and uses 100 samples from the posterior distribution.

**Hamiltonian Monte Carlo.** An MLP of 10 hidden layers with 300 neurons each is used as the underlying model. The sampling step consists of 70 samples with a sampling step of 0.00015. The first 10 samples are always burnt. A $\tau$ value of 1 is used. At inference time, we sample 100 times from the posterior distribution and drop the first 20. The model is trained for 50 epochs.

**Heteroscedastic Gaussian Process.** The implementation is the same as in the synthetic data case. The number of inducing points (see (Hastie et al., 2009) for an explanation) is set to 256, the learning rate to 0.0009, epochs to 50.

### C.4. Synthetic data experiments - Classification

To illustrate our arguments in the classification case, we make use of a binary classification setting with a data-generating process where $y_i \sim \text{Bern}(p(x_i))$ and $p(y_i|x_i) = 1$ is given by

$$p(y_i|x_i) = 0.5 \times \left( \sin\left( \frac{1}{x_i^3} \right) + 1 \right).$$

For this problem, we consider a multi-layer neural network that outputs $\hat{p}(y|x)$ and learns it using the binary cross-entropy loss.

### C.4.1. EXPERIMENTAL ILLUSTRATION

To evaluate the uncertainty quantification methods in this setting, we showcase the behavior of the Bayesian and ensemble approaches. We make use of the information theory result, where total uncertainty can be decomposed into conditional entropy and mutual information as follows

$$\underbrace{H(\mathbb{E}_\theta(\hat{p}(y_i|x_i)))}_{\text{Total Uncertainty}} = \underbrace{\mathbb{E}_\theta(H(\hat{p}(y_i|x_i, \theta)))}_{\text{Aleatoric Uncertainty}} + \underbrace{I(y_i, \theta)}_{\text{Epistemic Uncertainty}} \tag{11}$$

Figure 11 shows how the models are overconfident in regions where the bias is high and interpret the uncertainty as aleatoric uncertainty. This mirrors precisely the behavior observed in the regression setting: in regions where the predicted probability $\hat{p}(y_i \mid x_i)$ deviates systematically from the true conditional $p(y_i \mid x_i)$, the conditional entropy term $\mathbb{E}_\theta[H(\hat{p}(y_i \mid x_i, \theta))]$ is inflated, while the mutual information term $I(y_i, \theta)$ remains small. In other words, the systematic deviation of the mean

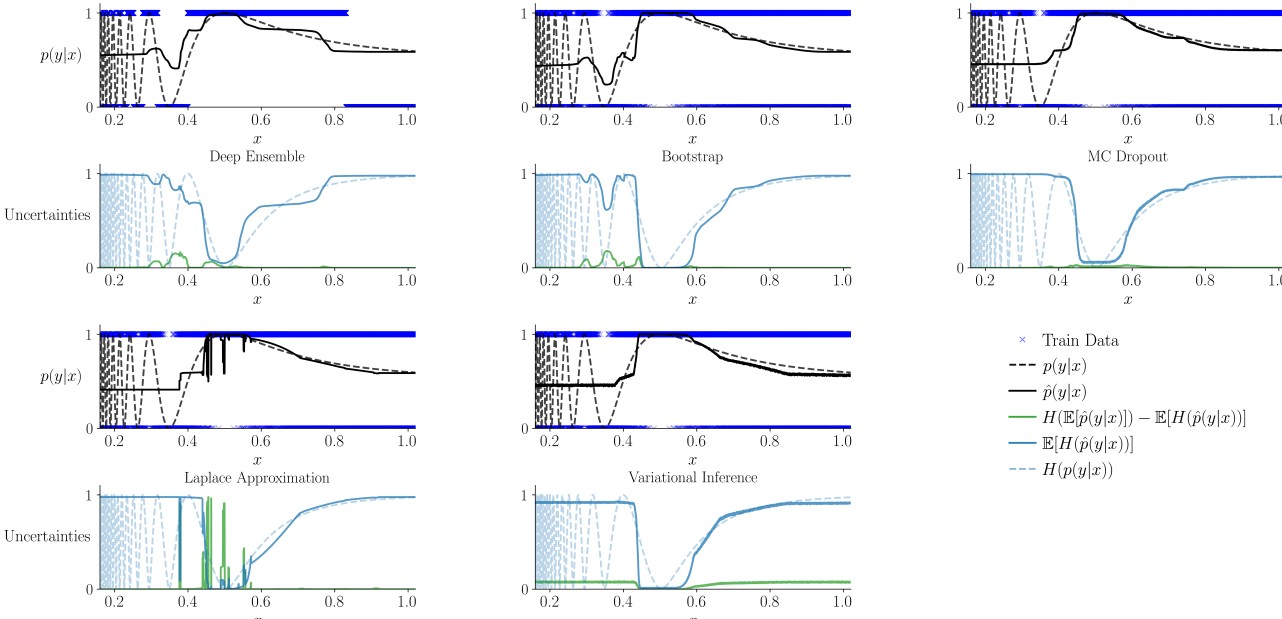

*Figure 11.* There are two panels for each model type. The panel on top shows the training data as blue marks, the true probability of $y$ given $x$, and the models' predicted probabilities. The panel on the bottom shows the aleatoric and epistemic uncertainty estimates, and the true aleatoric uncertainty (entropy of the true conditional probability $p(y \mid x)$). We see that when the model bias is high, aleatoric uncertainty is inflated and epistemic uncertainty is underestimated.

predictor is again absorbed into the aleatoric component, this time through the entropy of a misplaced Bernoulli mean rather than through an inflated predictive variance. Table 3 supports this observation: the column $\mathbb{E}_\theta[H(\hat{p}(y_i \mid x_i, \theta))] - H(p(y_i))$ is positive for every method, and its magnitude tends to track the prediction bias $\mathbb{E}_{\mathcal{D}_N}[\hat{p}(y \mid x) - p(y_i)]$. The estimated epistemic component, given by the mutual information column, is by comparison an order of magnitude smaller and does not increase to compensate. The bias that should have been recognized as a reducible, model-related (i.e. epistemic) deficiency is therefore systematically misattributed to irreducible noise, exactly as in the regression experiments.

Finally, the data-procedural decomposition of Sec. 3.2 also holds in the classification case. The mutual information $I(y_i, \theta)$ is computed from a second-order distribution whose members are trained on a single fixed dataset $\mathcal{D}_N$, so it can only reflect procedural variability $\mathbb{E}_{\mathcal{D}_N}[\mathrm{Var}_\gamma(\cdot) \mid \mathcal{D}_N]$ and not the data-driven term $\mathrm{Var}_{\mathcal{D}_N}(\mathbb{E}_\gamma[\cdot] \mid \mathcal{D}_N)$. We therefore expect the same systematic underestimation relative to a reference distribution that we documented for regression in Figures 9 and 10, confirming that neither failure mode is an artifact of the regression loss but rather a structural property of how aleatoric and epistemic uncertainty are decomposed under a second-order representation.

| Model | $\mathbb{E}_\theta(H(\hat{p}(y_i|x_i,\theta)))$ | $H(\mathbb{E}_\theta(\hat{p}(y_i|x_i)))$ $-\mathbb{E}_\theta(H(\hat{p}(y_i|x_i,\theta)))$ | $\mathbb{E}_{\mathcal{D}_N}[\hat{p}(y|x) - p(y_i)]$ | $\mathbb{E}_\theta(H(\hat{p}(y_i|x_i,\theta)))$ $-H(p(y_i))$ |
|---|---|---|---|---|
| Bootstrap | 0.756 | 0.012 | -0.003 | 0.064 |
| DE | 0.791 | 0.013 | 0.025 | 0.099 |
| VI | 0.736 | 0.06 | -0.013 | 0.044 |
| MC Dropout | 0.799 | 0.006 | 0.002 | 0.107 |
| Laplace | 0.777 | 0.019 | -0.011 | 0.086 |

*Table 3.* N=1000

### C.4.2. IMPLEMENTATION DETAILS BY METHOD

All methods are evaluated for different training dataset sizes: $\{50, 100, 500\}$ with batch sizes $\{32, 32, 64\}$ respectively. The experiments are repeated for 5 different seeds: 7, 42, 123, 999 and 2024. To conduct the experiments, we tuned all the

hyperparameters tuning over relevant parameters was performed.

**Deep Ensemble.** Trained with an MLP of 4 hidden layers with 100 neurons each, learning rate of 0.009. Training epochs: 250, 500, 1000. Every member of the deep ensemble uses the whole dataset.

**Bootstrap Ensemble.** Trained as the Deep Ensemble, but each ensemble member uses a boostrap sample with 60% of the training data.

**Monte Carlo Dropout.** A single MLP of 4 hidden layers with 100 neurons each is trained. We use a dropout rate of 0.2 at training and inference time, and a learning rate of 0.009.

**Variational Inference.** This model is trained based on Blundell et al. (2015). An MLP of 4 hidden layers with 100 neurons each converted to a Bayesian neural network is used. The model is trained with a learning rate of 0.009, 200 burn-in epochs, a beta value of 5, 10 Monte Carlo samples at training time, and 500 at test. The model is evaluated for 500 and 1000 training epochs.

**Laplace Approximation.** An MLP of 4 hidden layers with 100 neurons each is used to find the MAP estimate, together with a learning rate of 0.009. For the Laplace step, a prior precision of 1 and a noise value of 0.1 are used. The model is trained for 400, and 1000 epochs, and uses 1000 samples from the posterior distribution.

### C.5. Further figures

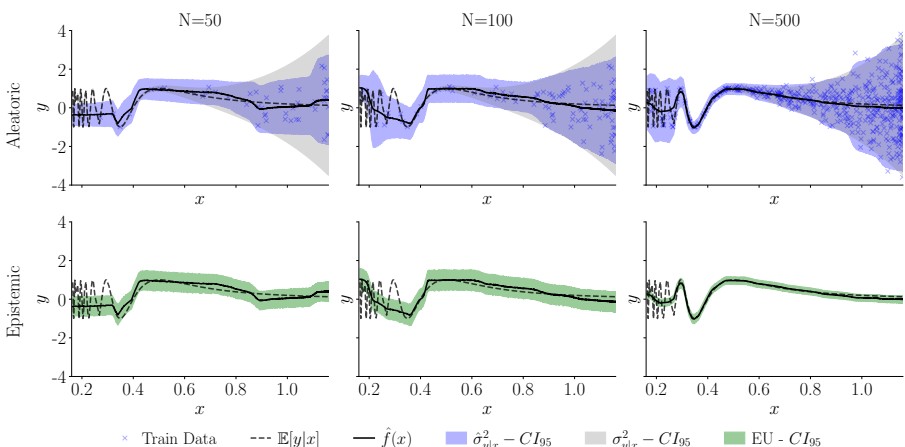

*Figure 12.* VI estimates, obtained with different sample sizes as in Figure 6.

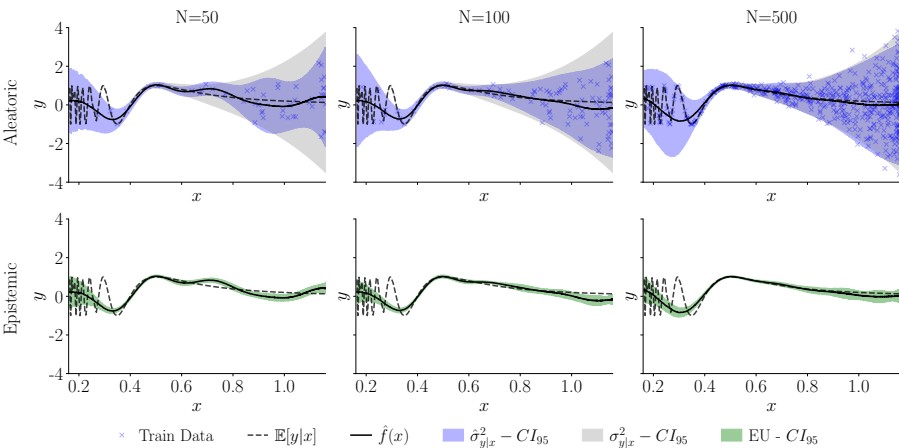

*Figure 13.* GP estimates, obtained with different sample sizes as in Figure 6.

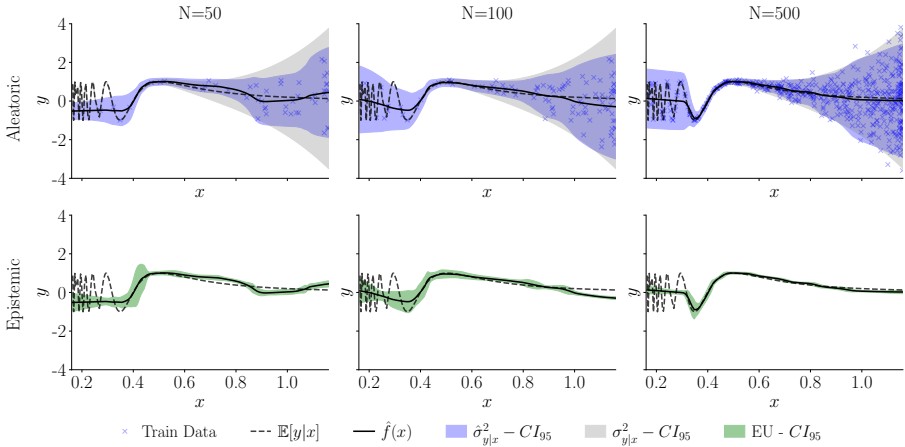

*Figure 14.* Bootstraped ensemble estimates, obtained with different sample sizes as in Figure 6.

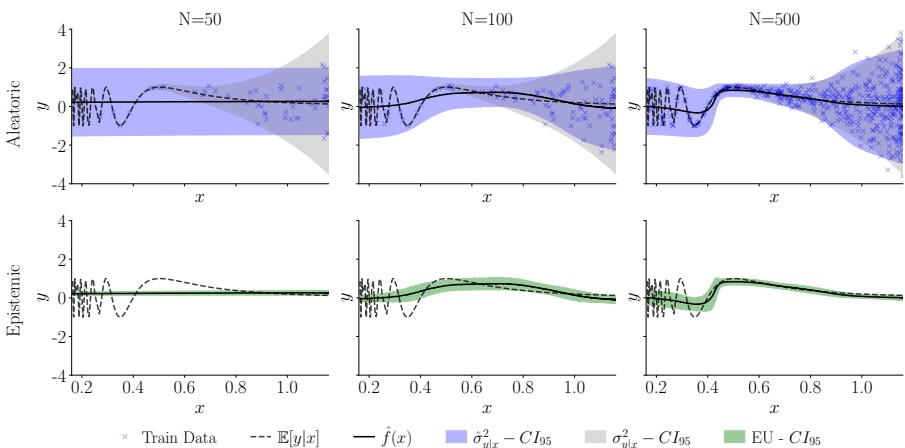

*Figure 15.* MC Dropout estimates, obtained with different sample sizes as in Figure 6.

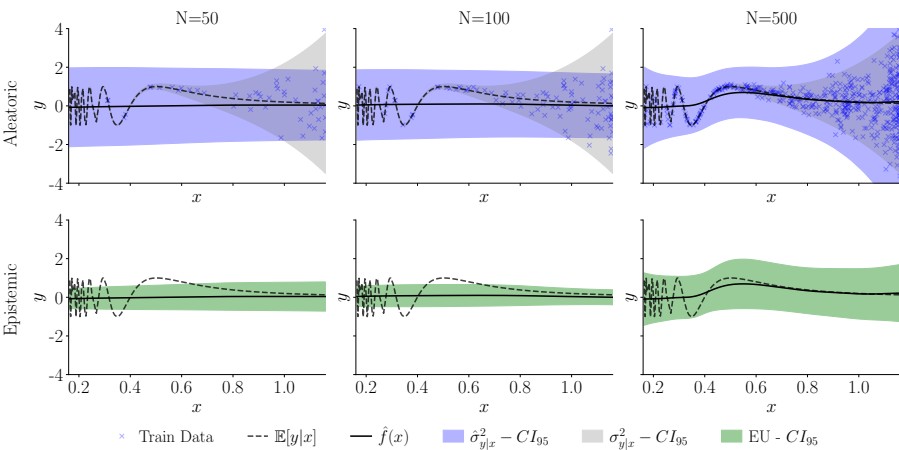

*Figure 16.* HMC estimates, obtained with different sample sizes as in Figure 6.

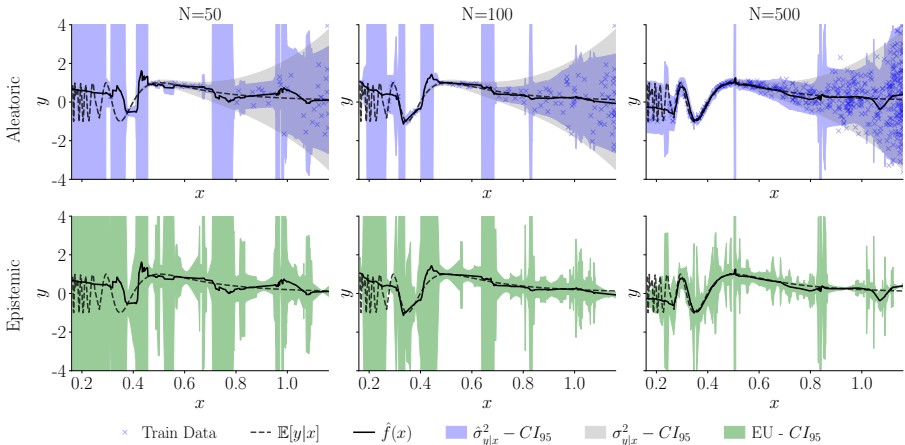

*Figure 17.* Laplace estimates, obtained with different sample sizes as in Figure 6.

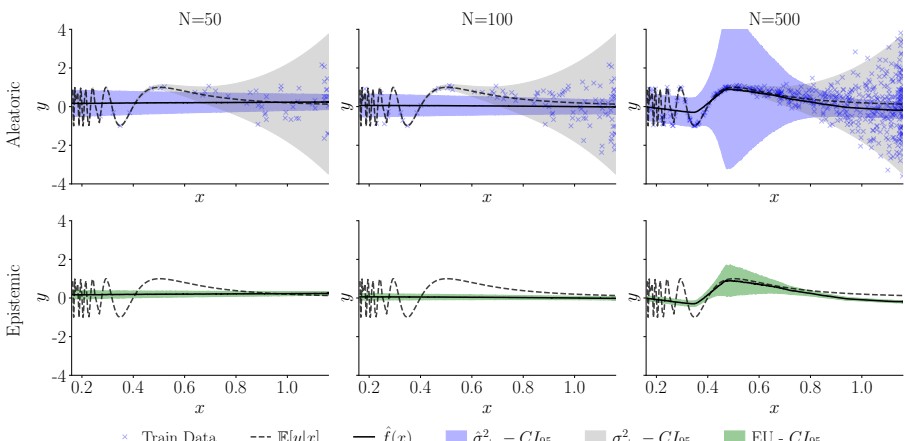

*Figure 18.* DER estimates, obtained with different sample sizes as in Figure 6.

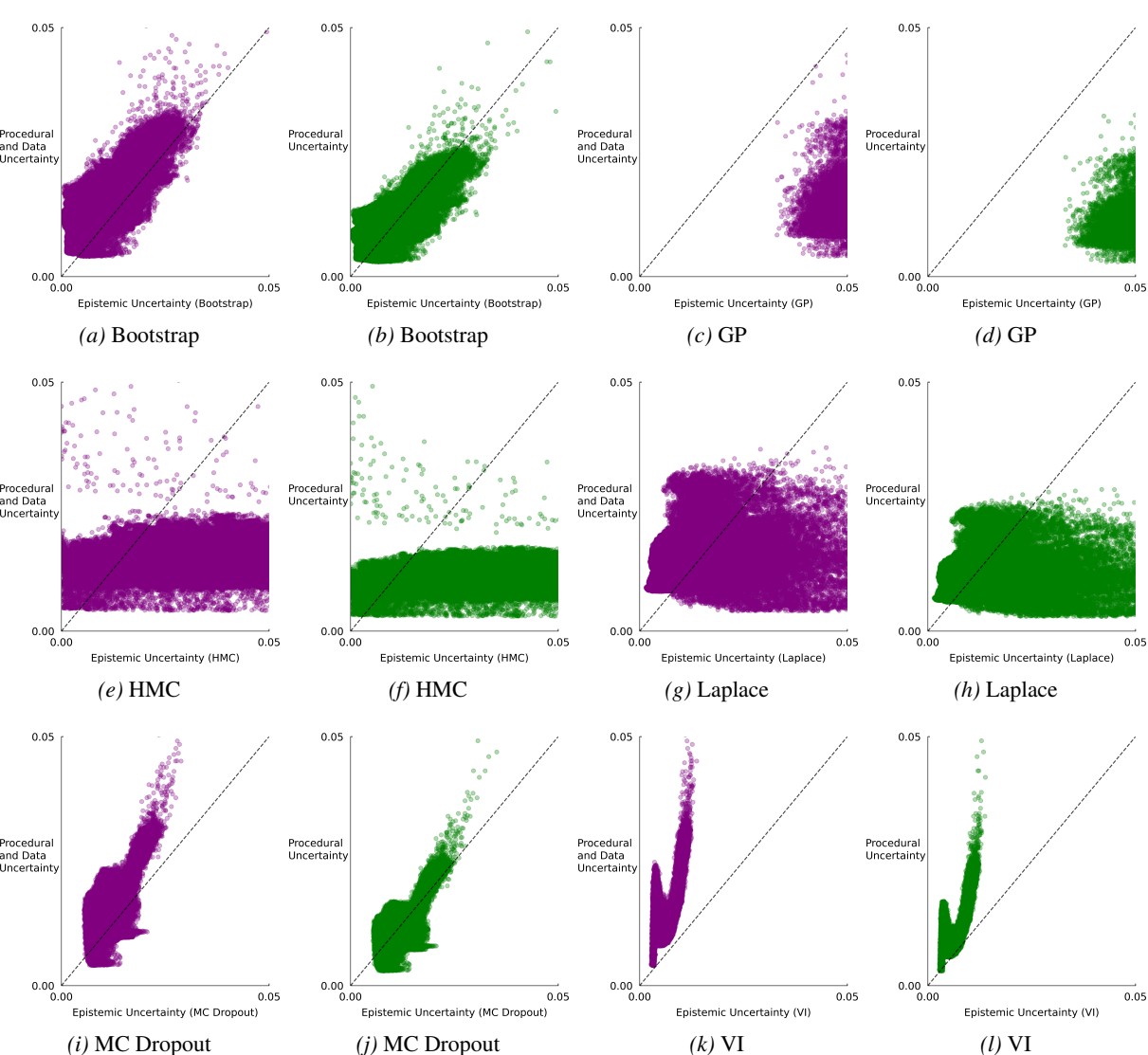

*Figure 19.* Comparison of procedural and data uncertainties vs epistemic uncertainty across methods on the Taxi dataset. Figures in purple (first figure of every method) show the relation between the epistemic uncertainty from the reference distribution (including procedural and data uncertainties), and green pictures (second figure of every method) show the relationship between the procedural uncertainty from the reference distribution and the epistemic uncertainty from each method. In general, we see that adding the data uncertainty component shifts the relation between the two estimates.

