# OpenReview forum: "Position: Epistemic Uncertainty Estimation Methods are Fundamentally Incomplete"
_ICML.cc/2026/Position_Paper_Track — ICML 2026 Position Paper Track regular_

### Official Review · Reviewer_f6SM · 2026-02-23

**Significance:** 4
**Argument Clarity:** 4
**Rating:** 6
**Confidence:** 4

**Questions:**

1. Do you have formal results showing that second-order distributions cannot (in a mathematical sense) represent bias or data uncertainty, or does "fundamental" refer to limitations inherent in current algorithmic implementations? If the latter, would a hypothetical bias-corrected ensemble that explicitly accounts for regularization-induced bias be considered "complete" under your framework?

2. While your theoretical framework encompasses both classification and regression (Section 2), the empirical validation focuses exclusively on heteroscedastic regression. Information-theoretic decompositions in classification (using entropy and mutual information) differ substantially from variance-based regression decompositions. Do you have empirical evidence or theoretical arguments demonstrating that the same bias contamination occurs in classification (e.g., Dirichlet-based evidential learning or Bayesian neural networks with categorical outputs)? If not, does this limit the universality of your claim that all second-order methods are fundamentally incomplete?

3. Your proposed evaluation standard requires either access to the true data-generating process (for synthetic resampling) or massive datasets enabling disjoint partitioning (e.g., 1.2M samples for the taxi experiments). For applications where uncertainty quantification is most critical but data is scarce (e.g., n < 1,000 in medical imaging or scientific experiments), computing reference distributions is infeasible. What evaluation protocols do you recommend for small-data regimes, or is your position that uncertainty estimates in such settings are inherently unverifiable and thus unsuitable for safety-critical deployment?

4. In Section 4.1, you characterize conformal prediction (CP) as addressing a "different problem" because it does not decompose uncertainty into aleatoric/epistemic components. However, CP provides finite-sample coverage guarantees regardless of model bias. Given that CP sidesteps the decomposition entirely, do you view it as a methodologically sound alternative that avoids the incompleteness you identify, or does CP suffer from similar blind spots regarding bias and variance? Specifically, does marginal or conditional coverage adequately address the systematic overconfidence issues you demonstrate in Figure 3?

**Alternative Views Section:**

Yes

**Compliance With Llm Reviewing Policy A Conservative:**

Affirmed.

**Discussion Potential:**

3

**Final Justification:**

The authors have provided very good answers that clarified my concerns

**Paper Summary:**

This paper argues that widely-used second-order methods for disentangling aleatoric and epistemic uncertainty in supervised learning are fundamentally incomplete. The authors advance two main claims to support this position. First, they demonstrate that unaccounted model bias contaminates uncertainty estimates by inflating aleatoric uncertainty estimates while deflating epistemic uncertainty estimates. Second, they show that existing methods capture only partial contributions to the variance-driven component of epistemic uncertainty, typically quantifying procedural uncertainty (variation from random initialization) while neglecting data uncertainty (variation across different training datasets). The paper introduces the concept of a "reference distribution" to empirically validate uncertainty estimates by decomposing variance into its data and procedural components. It includes a dedicated section on alternative views, discussing credible competing frameworks including Bayesian neural networks, deterministic uncertainty methods, credal sets, and conformal prediction. The authors conclude with a call to action advocating for: (1) a finer-grained taxonomy distinguishing model uncertainty (bias and approximation error) from estimation uncertainty (data and procedural components); (2) evaluation of new methods against reference distributions rather than only downstream tasks; and (3) treating epistemic uncertainty estimates as lower bounds in safety-critical applications while separately assessing model bias.

**Position:**

Yes

**Position In Title:**

Yes

**Related Work:**

3

**Strengths And Weaknesses:**

Strengths

The paper grounds its position in solid statistical theory, employing Bregman divergences and generalized bias-variance decompositions to formalize its arguments. By distinguishing five distinct sources of uncertainty (approximation error, estimation bias, data uncertainty, procedural uncertainty, and distributional uncertainty) in Figure 2, the authors provide a nuanced conceptual taxonomy that advances beyond prior binary classifications. This theoretical sophistication lends credibility to the claim that current methods are incomplete.
The position is supported by extensive experiments across both synthetic and real-world datasets (taxi trip duration), evaluating eight distinct methods including Bayesian approaches (VI, Laplace, HMC), ensembles (Deep Ensembles, Bootstrap), and deterministic approaches. The introduction of the "reference distribution" as an empirical benchmark to disentangle data and procedural uncertainty is methodologically innovative and provides tangible evidence for the paper's claims. The visual comparisons (Figures 3, 4, 7, 9) compellingly demonstrate that existing methods systematically underestimate variance and conflate bias with aleatoric noise.

Weaknesses

1. While the theoretical discussion encompasses both classification and regression, the empirical validation focuses almost exclusively on heteroscedastic regression. Given that information-theoretic decompositions in classification differ significantly from variance-based regression decompositions, the claim of fundamental incompleteness would be strengthened by including classification experiments (e.g., CIFAR-10/100 or ImageNet) to demonstrate the breadth of the critique.

2. The reference distribution proposal requires either access to the true data-generating process (for synthetic resampling) or massive datasets allowing disjoint partitioning (1.2M samples for the taxi dataset). The paper acknowledges that "sampling from the data-generating distribution is infeasible" for most real-world applications, yet this constraint significantly limits the practical utility of the proposed evaluation standard. The paper could better address how researchers should proceed when reference distributions cannot be computed.

**Support:**

4

---

> ### Author Rebuttal · Authors · 2026-03-30
>
> We thank the reviewer for the very positive feedback and constructive comments.
>
> **Q1: Meaning of the word “fundamental”, possible “completeness of a bias-corrected ensemble”.**
> The word “fundamental” refers to a structural limitation that is not specific for a single algorithm or method, but due to the discrepancy of aleatoric and epistemic uncertainty resulting from the bias-variance decomposition and disentangled estimates of existing second-order methods. We can make this more precise as follows. Under standard, unregularized empirical risk minimization and universal approximation, the dual mean $\bar{p}^\*$ converges to the Bayes optimal estimator $\hat{p}^*$ as $N\to\infty$, so the estimation bias vanishes asymptotically. However, when we introduce regularization (weight decay, dropout, early stopping), the estimator converges to a regularization-dependent limit that is not equal to the Bayes optimal estimator. Therefore, the estimation bias does not vanish with more data. The variance-based decomposition in Eqn. (3), which is used by second-order methods to disentangle uncertainties, has no term that can represent this positive bias.
>
> Bias-corrected ensembles could, in principle, mitigate this limitation. One example of related attempts is Gupta et al. (Ensembles of Classifiers: a Bias-Variance Perspective, TMLR 2022), who analyze bias and variance in classifier ensembles via bootstrap-based estimators. However, to truly estimate bias, one would require access to the ground truth conditional distribution, which is needed for the reference distribution in Definition 3.1. This is not feasible in real-world settings, which is why we argue that these methods are fundamentally incomplete. That said, we do believe that approximate bias-correction strategies (e.g., by resampling techniques, as in the paper cited above) represent a promising direction. We will expand on this discussion in the revised version of the text, and we thank the reviewer for the comment, as this encompasses ongoing directions of our own current work.
>
> **Q2: Evidence for the same bias contamination in the classification case.**
> Let us emphasize that the generalized bias-variance decomposition in Section 3.1 holds for any proper scoring rule. So, it also holds for classical loss functions used in classification settings, such as binary or multi-class cross-entropy. In the paper, we chose to focus on the regression setting, for which it is easier to visualize the key findings. We confirm that we observe exactly the same key findings in classification experiments (i.e., estimation bias and approximation distort aleatoric uncertainty estimates, and existing epistemic uncertainty estimators are incomplete because they do not represent data uncertainty and procedural uncertainty at the same time).
> In the revised version, we will include additional results for a classification problem, where we learn the probability function $p(y \mid x) = 0.5 \times (\sin(1/(5x^3)) + 1)$, which allows us to differentiate between high aleatoric and high epistemic uncertainty regions. These estimates are obtained using the information-theoretic decomposition. For a snapshot of these results, please see [this link](https://project-f9eox.vercel.app/).
>
>
> **Q3: Evaluation protocol for small-data regimes, and benchmarks such as CIFAR-10 or Imagenet.**
> See response to question 3 of reviewer imZM.
>
>
> **Q4: CP as an alternative.**
> We do agree with the reviewer that CP provides finite-sample coverage guarantees regardless of bias. As such, it does not suffer from the issues we identify in our position paper. Generally speaking, what typically happens with CP is that prediction sets become smaller when models become more accurate. So, higher epistemic or aleatoric uncertainty at train time will result in smaller prediction sets. In split CP, high epistemic uncertainty for the calibration dataset (i.e., a small calibration dataset) will result in bigger deviations from the desired coverage level on a single testset.
>
> However, classical CP methods do not have the intention to disentangle aleatoric and epistemic uncertainty. They intend to model total uncertainty, which is the combination of aleatoric and epistemic uncertainty. Combining CP with second-order models, as done in several recent papers, can result in disentanglement and coverage guarantees at the same time. We will update the Alternative Views section to stress this point.
>
> Refs.
> - Rossellini et al. Integrating uncertainty awareness into conformalized quantile regression. AISTATS 2024.
> - Azizi et al.  CLEAR: Calibrated learning for epistemic and aleatoric risk. ICLR 2026.
> - Karimi and Samavi. Evidential uncertainty sets in deep classifiers using conformal prediction. COPA 2024.
> - Cabezas et al. Epistemic uncertainty in conformal scores: A unified approach. arXiv preprint arXiv:2502.06995, 2025.
> - Caprio et al. Conformalized credal regions for classification with ambiguous ground truth, TMLR 2025.

---

> > ### Author Rebuttal · Reviewer_f6SM · 2026-04-01
> >
> > Very good answers that clarify my concerns

---

### Official Review · Reviewer_o1CV · 2026-03-08

**Significance:** 2
**Argument Clarity:** 2
**Rating:** 4
**Confidence:** 2

**Questions:**

- Related to the last weakness above, I’m curios what do the authors think about the “usefulness” of uncertainty decomposition? Based on your work it seems that “proper” decomposition is quite hard and non-trivial, so I wonder if we should even strive for it or should just aim for a single, well-calibrated confidence score instead? This question is not related to my assessment of the paper, but I’m just interested in hearing authors’ thoughts on this matter

**Alternative Views Section:**

Yes

**Compliance With Llm Reviewing Policy A Conservative:**

Affirmed.

**Discussion Potential:**

2

**Final Justification:**

I trust the authors they will indeed add a proper **Alternative Views** section, which was my main concern. And some more discussion on the motivation/practicality of EU vs AU decomposition. It would be good to start questioning this more, as it's the only way for such ideas to be adopted beyond the "theory UQ bubble" in my opinion

Though my concern on this being a position paper submission (instead of being a main track submission) still remains. Though it's my first time reviewing position papers, so if this concern is not shared by other reviewers/AC, then it's just my lack of experience with such papers

**Paper Summary:**

The submitted manuscript discusses uncertainty decomposition into aleatoric and epistemic parts for 1-dimensional multi-variate regression problems. The main claim of the paper is that the usual regression decomposition is flawed since it i) overestimates the aleatoric uncertainty while at the same time underestimates the epistemic uncertainty due to bias and ii) captures only procedural uncertainty (i.e., uncertainty when varying model/training hyperparameters) while ignoring data uncertainty (i.e., uncertainty when training on different realizations of the training dataset).

While I find some ideas in the paper interesting, I am not sure the paper fits the criteria of a position track paper (though it's my first time reviewing for position track): to me the paper reads more like a main track paper where the technical contributions are the two aforementioned ideas around the uncertainty decomposition. Also the Alternative Views section is entirely "off" in my opinion, it reads like a background section on existing uncertainty quantification approaches and does not discuss the alternative/opposing views to the arguments presented in the paper.

**Position:**

Yes

**Position In Title:**

No

**Related Work:**

1

**Strengths And Weaknesses:**

Strengths:
- I like Figure 2 and the decomposition of epistemic uncertainty into 5 different sources. This very nicely illustrates how “overloaded” the term epistemic uncertainty is
- Some of the theoretical arguments are interesting and backed by experiments, so I think the paper has potential as a main-track submission

Weaknesses:
- Since all of the examples and experiments deal with 1-dimensional regression, I think it would be better to reflect this in the title and abstract already.
- Section 2 could be organised a bit better I think - for example, \tilde{p} is used in Eq. (4) while then only being defined a couple of paragraphs later
- As mentioned above, the Alternative Views section reads like a background section and I'm not entirely sure how it relates to the decomposition arguments presented in the rest of the paper?
- While I like the idea of using a reference distribution to capture also the data uncertainty, I have some concerns about the practicality/feasibility of the proposed approach to fit n_{\gamma} x n_{d} different models
- Given there’s been a lot of recent work that questioned epistemic vs aleatoric decomposition (e.g., [1]), the paper would benefit greatly from a more in-depth discussion of how the submitted paper relates to prior work. If you stay with the position paper format, maybe you could use the Alternative Views section for that?
-  The paper often says that decomposing uncertainty into aleatoric and epistemic is important for safety-critical applications, but it fails to provide concrete examples. Is active learning the only one? I feel like this would be important especially given that the authors want to frame this work as a position paper

[1] Smith et al. Rethinking Aleatoric and Epistemic Uncertainty

**Support:**

3

---

> ### Author Rebuttal · Authors · 2026-03-30
>
> We thank the reviewer for the constructive feedback and helpful comments.
>
> **Comment: Suitability as a position paper.**
> In our opinion, the paper fits in the position paper track, because it takes a clear, argumentative position about a widely-used group of ML methods. The experiments in our submission are meant to provide additional intuition and insight to support the position that we take. This is not uncommon in the ICML position paper track. Here are some examples of accepted ICML position track papers in 2024 or 2025 that include experimental results to support their position:
>
> - Supervised Classifiers Answer the Wrong Question for OOD Detection (ICML 2025).
> - Theory of Mind Benchmarks are Broken for Large Language Models (ICML 2025).
> - Graph Matching Systems Deserve Better Benchmarks (ICML 2025).
> - Benchmarking is Limited in Reinforcement Learning Research (ICML 2024).
>
> **Q1: 1-dimensional regression setup.**
> The 1-feature regression experiment serves as the running example throughout the paper, because it allows us to illustrate the core arguments transparently and visually. However, we note that the paper already includes experiments on a multidimensional regression task (Appendix C.3). We will make this point more prominent in the main text. Additionally, we will include results for binary classification (see the response to Q2 from Reviewer f6SM), demonstrating the generality of our position beyond the regression case.
>
> **Q2: Fix of notation for Section 2.**
> See response to Q2 of reviewer imZm.
>
> **Q3: Alternative Views.**
> We agree that the current Alternative Views section is written a bit too survey-like and does not yet sufficiently present the opposing views on the topic. In revision, we therefore plan to reorganise it around explicit alternative views, while retaining much of the current content:
>
> - The current Section 4.1 will be reframed as the alternative view that standard second-order methods already provide suitable proxies for aleatoric and epistemic uncertainty. This view is implicit in much of the literature on second-order methods, which is why the existing discussion on them will remain. We plan to include the current review as representative literature embodying this view. We will then state our disagreement more explicitly by linking back to Section 3.1. and Section 3.2.
> - The subsequent subsection will discuss and critically analyse the opposing view that downstream-task performance on OOD detection, selective prediction and active learning is the appropriate way to evaluate uncertainty methods. The current discussion of the limitations of these performance metrics already appears at the end of Section 3.2. In revision, we will make this argument a dedicated subsection of Section 4. We will clarify that downstream performance can be highly relevant, but that it does not lead to guarantees about the faithfulness of the uncertainty estimates.
> - The third subsection will discuss recent critical works on the aleatoric/epistemic uncertainty decomposition itself, and relate our work to it. We already cite this literature in the introduction, but the revised Section 4 will discuss it directly. Recent other critiques such as Wimmer at al. [2023], Muscányi et al [2024] and Smith et al. [2025] represent the alternative view that the standard aleatoric/epistemic decomposition can be conceptually flawed, empirically correlated or too coarse. Our view overlaps with this line of research, but instead of discarding the decomposition in total, we argue for refining it into a more granular taxonomy.
> - The current Section 4.2 on deterministic methods, credal sets and conformal prediction will be retained, but reformulated more explicitly around the question of what uncertainty objects these methods represent. This will make clear why conformal prediction is used to fulfill coverage guarantees rather than represent aleatoric and epistemic uncertainty, while credal or deterministic approaches may be valuable alternatives for certain problems without being substitutes for the notion of epistemic uncertainty as studied in Sections 3.1 and 3.2.
>
> **Q4: Practicality of using a reference distribution.**
> See answer reviewer imZM.
>
> **Q5: Prior work.**
> See answer to Q3.
>
> **Q6: Importance of disentanglement.**
> Active learning is only one example. If uncertainty disentanglement works well, it has the potential to explain to end users when and why an ML fails to provide accurate predictions. The appropriate intervention then differs for the different types of uncertainty. High aleatoric uncertainty supports actions like abstaining from prediction or adding informative features, whereas high epistemic uncertainty supports collecting more representative training data. This source-specific actionability is exactly why disentanglement of uncertainty matters in applications where wrong predictions have a strong impact. We will add a short paragraph in the introduction to make this motivation explicit.

---

> > ### Author Rebuttal · Reviewer_o1CV · 2026-04-02
> >
> > Thanks for the response. I trust the authors they will indeed add a proper **Alternative Views** section, which was my main concern. Perhaps others could include there also some discussion on the importance/practicality of the EU vs AU decomposition (from our discussion above), it would be good to start to at least questioning this as an UQ community in my opinion

---

### Official Review · Reviewer_RsBB · 2026-03-11

**Significance:** 3
**Argument Clarity:** 3
**Rating:** 5
**Confidence:** 4

**Questions:**

I have no further questions.

**Alternative Views Section:**

Yes

**Compliance With Llm Reviewing Policy A Conservative:**

Affirmed.

**Discussion Potential:**

3

**Final Justification:**

Given that I only saw minor weaknesses, and the responses the authors provided to the other reviewers I keep recommending acceptance.

**Paper Summary:**

The authors argue that current methods of disentangling predictive uncertainty into aleatoric and epistemic components are flawed and require a more fine-grained decomposition and analysis.

**Position:**

Yes

**Position In Title:**

Yes

**Related Work:**

3

**Strengths And Weaknesses:**

### Strengths
- The paper is well written and easy to follow, with a clearly defined and well-motivated position.
- How to represent, decompose, and even understand what uncertainty is is an ongoing discussion in the community.
- The proposed extension is clearly discussed and evaluated in the appendix using a wide series of methods on relatively small-scale benchmarks.

### Weaknesses
- The reference section needs to be cleaned up; e.g., Charpentier et al. (2021) is a published paper;
Charpentier et al. (2020), Daxberger et al. (2021), and Geifman et al. (2017) are all papers published at NeurIPS, yet their publication venue is written completely differently;
Ulmer et al. lacks a year, etc.
- `\citet` vs `\citep` are used inconsistently


Minor sidenote: _"Together, these results
highlight that current epistemic uncertainty es-
timates can only be used in safety-critical and
high-stakes decision-making when limitations are
fully understood by end users and acknowledged
by AI developers."_
-> I strongly hope this applies to any tool, be it ML-based or not, employed in such scenarios.

**Support:**

3

---

> ### Author Rebuttal · Authors · 2026-03-30
>
> We would like to thank the reviewer for the positive assessment and the constructive comments.
>
> **Q1: Bibliography and citation style.**
> We will ensure that all published papers cite the correct venue uniformly, and we will fix the missing year for Ulmer et al. We will also fix the use of \citet and \citep throughout the paper.
>
> **Q2: Minor sidenote.**
> We fully agree with the reviewer: the principle that limitations must be understood before deploying any tool in safety-critical settings is universal and not specific to ML-based uncertainty methods. Our intent was to emphasize this point within the uncertainty estimation community, where the gap between what methods claim to measure and what they actually capture is, in our view, insufficiently acknowledged. With this work, we hope to contribute to a more critical and transparent development of uncertainty quantification methods.

---

> > ### Author Rebuttal · Reviewer_RsBB · 2026-04-02
> >
> > Thank you for your rebuttal.

---

### Official Review · Reviewer_imZM · 2026-03-12

**Significance:** 2
**Argument Clarity:** 3
**Rating:** 4
**Confidence:** 1

**Questions:**

- Line 145, right column. Missing period.
- Line 171, left column. The definition of procedural uncertainty is not consistent with Figure 2.
- In order to genuinely evaluate the decomposition of epistemic and aleatoric uncertainties, we might have to resort to reference distribution. As the authors mentioned, the reference distribution can only be estimated with random datasets and random procedure samples in practice. Doesn't it introduce uncertainties themselves?
- In Call to Action, the authors suggest treating epistemic uncertainty estimates as **lower** bounds on reducible uncertainty. What is the meaningful **upper** bound of reducible uncertainty instead?

**Alternative Views Section:**

Yes

**Compliance With Llm Reviewing Policy A Conservative:**

Affirmed.

**Discussion Potential:**

2

**Final Justification:**

I have read the rebuttal and other reviews. The idea is fine and likely to raise discussion in the AI community. But the alternative views section as raised by Reviewer o1CV should be reorganized to make more discussions on the connection with related work. Considering this, I would vote for weak accept but with low confidence since I am not quite familiar with this area.

**Paper Summary:**

This paper presents a position that existing commonly used epistemic uncertainty estimation methods are fundamentally incomplete. The authors show that bias estimation distorts both aleatoric and epistemic uncertainty estimates, with a general paradigm to identify sources of epistemic uncertainty. Extensive experimental results are conducted and alternative views are discussed.

**Position:**

Yes

**Position In Title:**

Yes

**Related Work:**

3

**Strengths And Weaknesses:**

**Strengths**
- The studied problem is fundamental for all data-driven learning methods, and relevant and important to the ICML community.
- Extensive experimental results on both simulated and real-world datasets are conducted to support some claims related to the proposed position.


**Weaknesses**
- I don't see obvious weaknesses in this manuscript.

**Support:**

3

---

> ### Author Rebuttal · Authors · 2026-03-30
>
> We thank the reviewer for the positive feedback and the constructive comments. We will address the open points one by one.
>
> **Q1: Issue in line 145.** Solved.
>
> **Q2: The definition of procedural uncertainty is not consistent with Figure 2.**
> We thank the reviewer for pointing this out, and agree that the current wording is not fully consistent with Figure 2. In the text, we currently make use of $\hat{p}\_{\gamma}(y|x)$, which corresponds to conditioning on $\gamma$ and is not the object shown in the figure. We will therefore revise the paragraph as follows: We define procedural uncertainty as the discrepancy between $\hat{p}\_{N, \gamma}(y|x)$ and $\hat{p}_N(y|x)$ and data uncertainty as the discrepancy between $\hat{p}_N(y|x)$ and the Bregman centroid $\bar{p}^*(y|x)$. We will remove $\hat{p}\_{\gamma}$ from the main text, which makes the notation consistent with Figure 2 and the later conditional variance decomposition in Eqn. (8).
> As a response to reviewer o1CV: we will replace $\tilde{p}$ by $p$ in Eqn. (4), as it is merely a variable that is minimized over. Hence, it does not need to be formally introduced, but we see that it creates confusion with the later introduced $\tilde{p}$.
>
> **Q3: Data-dependency in the estimate of the reference distribution.**
> Yes, in principle that is the case. However, on synthetic datasets, a sufficient number of samples can be drawn in the generation of training datasets and procedural parameters. The more samples are drawn, the better the reference distribution is estimated. So, uncertainty in the estimation of the reference distribution can be controlled.
> For real-world datasets, the quality of the estimate of the reference distribution largely depends on the size of the dataset that is used. That’s why we would only recommend this procedure for very large datasets that are not too high-dimensional, such as the taxi trip duration dataset that is used in our experiments. Standard multi-class classification benchmarks (e.g. CIFAR-10 or Imagenet) are typically less useful to estimate a reference distribution, because these datasets don’t have enough instances in relation to the number of classes. The size of the dataset will influence how accurately systematic bias, approximation error and data uncertainty can be estimated. For estimating procedural uncertainty, there is no substantial difference between synthetic and real-world datasets.
>
> Furthermore, we would like to clarify that this paper does not have the intention to propose a protocol for evaluating the trustworthiness of epistemic uncertainty on a specific dataset (e.g. a medical dataset). The ambition is rather to propose a protocol that can be used to evaluate methods that claim to be capable of measuring epistemic uncertainty. A key element of this evaluation protocol is a statistical simulation, where training data can be resampled. When newly-proposed methods already estimate epistemic uncertainty wrongly on simple toy problems with low-dimensional feature spaces, there is very little hope that such methods will behave correctly on more complex real-world datasets. With the taxi trip duration dataset, we illustrate that pathologies observed in simulations also appear in real-world datasets. However, our protocol is not meant to be used for benchmarking studies on specific real-world datasets.
>
> **Q4: Meaningful upper bound for reducible epistemic uncertainty.**
> We agree that the wording may have suggested the existence of an upper bound. However, our point is that current second-order epistemic estimates often capture only part of reducible uncertainty, since they may neglect both bias and data variability. Hence they should be interpreted as partial, rather than complete estimates. In general, we do not believe there is a model-agnostic upper bound on reducible uncertainty for a single dataset. Model-dependent bounds can be obtained under additional assumptions (e.g. a specified Bayesian priors), but these are assumption-dependent rather than generally meaningful guarantees. We will clarify this in the revision and revise the phrasing accordingly.

---

> > ### Author Rebuttal · Reviewer_imZM · 2026-04-01
> >
> > Thank the authors for the rebuttal. I will not change my score and still lean towards acceptance.

---

### Decision · Program_Chairs · 2026-04-30

**Decision:**

Accept (regular)

**Comment:**

Based on the reviews and rebuttal, I’m proposing accept for the paper.

There is a general consensus among the reviewers about the value of the paper. The reviewers appreciated the formal yet intuitive articulation of the problem together with simple empirical illustrations. In the first round of reviews a number of concerns were raised:

1. Position vs technical paper. Reviewers felt like the paper is written more as a main track contribution rather than a position paper and it may miss some of the aspects expected for a position paper. In the rebuttal, the authors sharpened the position (see response Q1 to f6SM) and comparison with alternative views (see response Q3 to Rev. o1CV and response Q4 to f6SM) and I believe these additional comments can be easily integrated in the camera ready version of the paper.

2. Limitations in the experiments. In the rebuttal, the authors covered classification to confirm the issues are not specific to regression and pointed out experiments already going beyond the 1d case. As for point 1, I strongly encourage the authors to integrate these aspects in the revised paper.

3. Small data regime. The authors acknowledged this aspect and clarified the need for larger datasets for their method. I think this is an important dimension (and possibly limitation) of the authors' proposal. Nonetheless, for a position paper, it does not compromise the value and interest of the paper.